# [Re] On the Reproducibility of Post-Hoc Concept Bottleneck Models

**Nesta Midavaine\*, Gregory Hok Tjoan Go\*, Diego Cánez Ildefonso\*, Ioana Simion\*, Satchit Chatterji**[†]

*{nesta.midavaine, gregory.go, diego.canez.ildefonso, ioana.simion, satchit.chatterji}@student.uva.nl*
*Graduate School of Informatics*
*University of Amsterdam*

**Reviewed on OpenReview:** *https://openreview.net/forum?id=8UfhCZjOV7*

## Abstract

To obtain state-of-the-art performance, many deeper artificial intelligence models sacrifice human explainability in their decision-making. One solution proposed for achieving top performance and retaining explainability is the Post-Hoc Concept Bottleneck Model (PCBM) (Yuksekgonul et al., 2023), which can convert the embeddings of any deep neural network into a set of human-interpretable concept weights. In this work, we reproduce and expand upon the findings of Yuksekgonul et al. (2023), showing that while their claims and results do generally hold, some of them could not be sufficiently replicated. Specifically, the claims relating to PCBM performance preservation and its non-requirement of labeled concept datasets were generally reproduced, whereas the one claiming its model editing capabilities was not. Beyond these results, our contributions to their work include evidence that PCBMs may work for audio classification problems, verification of the interpretability of their methods, and updates to their code for missing implementations. The code for our implementations can be found in https://github.com/dgcnz/FACT.

## 1 Introduction

There is an increasing demand within society to make artificially intelligent systems more explainable due to concerns of algorithmic bias (Pessach & Shmueli, 2023; Suresh & Guttag, 2021). One method of doing so involves Concept Bottleneck Models (CBMs), which train a model in an end-to-end fashion via concept prediction (Koh et al., 2020). These concepts are then used to predict the label attached to the data point, thus improving the model explainability and allowing for the detection of concept prediction-related mistakes through model-based interpretability. However, despite their benefits, CBMs have been widely criticized for being unable to deliver on its goals of interpretability, predictability, and intervenability (Raman et al., 2024; Havasi et al., 2022; Furby et al., 2024; Margeloiu et al., 2021). For example, Margeloiu et al. (2021) have found that when the concept predictor and classifier of a CBM are trained jointly, extra information about the targets is learned in the concept predictor. As such, interpretability and intervenability become lost as other factors beyond the concepts influence the decisions made.

By combining model-based and post-hoc paradigms, Yuksekgonul et al. (2023) proposed Post-hoc Concept Bottleneck Models (PCBMs). Through these, they address the CBM's limitations of data usage, decreased performance, and lack of model editing capabilities. The limitation of data usage is tackled by the implementation of multimodal models to obtain concepts automatically and decreased performance is addressed through the introduction of a residual modeling step.

This paper attempts to reproduce and extend upon their main findings through the following:

---

\*Equal contribution.
[†]Supervisor.

- **Reproducing their results using their provided codebase**. This was done to identify which of the results supporting their claims can be reproduced, and the resource costs involved (i.e., computational cost and development effort).

- **Reproducing their user study**. This relates to their claim stating that human-guided editing is fast and improves classification accuracy, which was supported using a survey with the participants being machine learning practitioners and researchers. We investigate this by replicating said survey with a different group of participants who come from a similar demographic to that of the original.

- **Verifying the interpretability of their approach**. The original paper assesses the interpretability of PCBMs by highlighting crucial concepts in specific classes for some of the used datasets. It also argues that the performance increase from model editing stems from the PCBM's interpretability. To inspect this, we examine two directions: First, we analyze what happens when the original authors' concept collection process is replaced with using meaningless concepts, which we then evaluate based on interpretability and performance. This serves as a baseline for our other interpretability experiments and also for the performance of PCBMs as it answers the question: *How much of the performance retained by the bottleneck can be attributed to the usefulness of concepts?*

  Furthermore, we evaluate the interpretability of PCBMs by examining if the concepts used by the model correctly correspond to objects in the input space. Similar to the work of Margeloiu et al. (2021) criticizing CBMs for not being truly interpretable, we utilize saliency maps (Simonyan et al., 2014; Smilkov et al., 2017) to visualize the correspondence between the concepts used and parts of the image. Additionally, we conduct another experiment to show the correspondence between concepts and the input space more generally for entire datasets instead of single images.

- **Extending their work through an additional experiment**. This experiment examines how the original implementation can extend to another modality, namely, audio. This was performed using AudioCLIP (Guzhov et al., 2022), which is a multimodal model we can utilize to automatically obtain concepts for audio. It was utilized in the same way (Yuksekgonul et al., 2023) did for images.

- **Improving the original code by implementing additional scripts**. This includes, for example, bash and python scripts which allow for easier reproduction, and master Jupyter Notebooks containing essentially all the setup and code needed to run the experiments.

After performing the above, we successfully verified their claims on model performance and the usability of CLIP concepts, though encountered issues replicating the claim on PCBMs enabling global model editing. In addition, we discovered limitations in the meaningfulness of concepts and their correspondence to objects in input images, and that PCBMs could be applied to inputs that are not in the form of image data.

As such, our replication efforts yielded mixed results due to potential implementation mistakes and incomplete experiment setup details. While some performance similarities were observed, inconsistencies emerged, particularly with the COCO-Stuff and SIIM-ISIC datasets. While PCBM-h's exhibited similar performance to the original model, the PCBMs showed mixed results. Also, our interpretability analysis revealed that despite performing well, random concepts lack interpretability. There were also limitations in interpretability due to concept feature values not being solely derived from the concept's presence in the image.

## 2  Scope of Reproducibility

PCBMs were introduced to imbue interpretability into any neural network while preserving both its performance and flexibility. They build upon the idea of concept analysis (Kim et al., 2018), which aims to comprehend how such networks generate and leverage high-level, human-understandable features. This inspired the development of Concept Activation Vectors (CAVs), forming the foundation of PCBMs. Yuksekgonul et al. (2023) proposed three claims in their paper which we aim to investigate:

1. **Claim 1: PCBMs achieve comparable performance to the original model.** The authors claim in their paper that after applying the Post-Hoc Concept Bottlenecks to various models, classification performance was comparable and in some cases even identical to the original baselines in

all scenarios. It must be noted, however, that this claim only holds for their hybrid implementation which they call "PCBM-h." An explanation for it can be found in subsection 3.1.

2. **Claim 2: PCBMs do not require labeled concept datasets.** Another claim made is that multimodal representations can be used in the absence of labeled data for concept generation. The original authors demonstrated this using CLIP for concept generation and showed that it improved the performance of the PCBM due to the concepts generated being more expressive.

3. **Claim 3: PCBMs allow for global model editing.** The claim states that by simply adjusting the concept weights, the PCBMs can be adapted to new distributions. In addition, it is shown through a user study that pruning the least human-logical concepts always results in improved classification performance.

# 3 Methodology

## 3.1 Model descriptions

The original paper introduces a post-hoc method aimed at enhancing the interpretability of neural networks. It demonstrates this approach using three different pre-trained backbone models: ResNet18 (He et al., 2016), CLIP-ResNet50 (Radford et al., 2021), and Inception (Szegedy et al., 2015). These models are freely available online, with the weights specifically for the Inception model trained on HAM10000 accessible from Daneshjou et al. (2022a). All our experiments involve classification tasks, focusing on the performance of PCBM and PCBM-h. As a baseline, we employ a linear probe at the output layer for model-dataset combinations lacking a classification head. This involves using logistic regression with $l_2$ regularization.

To convert the backbone model into a PCBM, the original authors projected the final embeddings of the backbone model onto a concept subspace. This projection is then used to train an interpretable linear classifier. The concept subspace is defined using a concept library which can be denoted as $I = \{i_1, i_2, \ldots, i_{N_c}\}$, where $N_c$ denotes the number of concepts. Each concept can be learned directly from the data or selected by a domain expert (Ghorbani et al., 2019; Yeh et al., 2020).

The original authors used two different methods to learn concept representations, with the first being through CAVs (Kim et al., 2018). To obtain a CAV $c_i$ for each concept $i$, we need two image embedding sets $P_i$ and $N_i$. The former comprises the embeddings of $N_p = 50$ images containing the concept, which we call the positive image examples $x_p$. Meanwhile, the latter comprises the embeddings of $N_n = 50$ random images not containing said concept which we refer to as the negative image examples $x_n$. This gives us $P_i = \{f(x_{p_1}), \ldots, f(x_{p_{N_p}})\}$ and $N_i = \{f(x_{n_1}), \ldots, f(x_{n_{N_n}})\}$, which we use to train a linear SVM that returns the CAV via its vector normal to the linear classification boundary. Note that obtaining these CAVs would require a densely annotated dataset with positive examples for each concept.

Meanwhile, the second approach involves utilizing the image and text encoders of a multimodal model (i.e., CLIP (Radford et al., 2021)) to generate embeddings for each modality. These encoders map to a shared embedding space, meaning that we can use the vector representations of the concept's natural language descriptions as the concept vectors. As such, we have for the concept "stripes" that $c_{stripes}^{text} = f_{text}(stripes)$. The vector representations of concepts are collected in a projection matrix $C \in \mathbb{R}^{N_c \times d}$, such that each row represents a concept vector of dimensionality $d$.

After we obtain $C$, the final embeddings of the backbone are projected onto the concept subspace. This matrix is then used to compute $f_C(x) = proj_C f(x) \in \mathbb{R}^{N_c}$ in a way such that $f_C^{(i)}(x) = \frac{f(x) \cdot c_i}{||c_i||_2^2}$. A way to interpret $f_C^{(i)}(x)$ is as the concept feature value of concept $i$ for image $x$, which we further examine in subsection 3.4. This means the concept feature value says something about the correspondence between concept $i$ and image $x$. Thus, the vector $f_C(x)$ can be used as the feature matrix of some interpretable model. Using this function, the original authors minimized the following loss for the computation:

$$\min_{g} \mathbb{E}_{(x,y)\sim D}[\mathcal{L}(g(f_C(x)), y)] + \frac{\lambda}{N_c K}\Omega(g), \tag{1}$$

where $\mathcal{L}$ represents the cross-entropy loss and the function $g$ represents a sparse linear model as determined by the authors, denoted as $g(f_C(x)) = f_C(x)\boldsymbol{W} + \boldsymbol{b}$. Here $\boldsymbol{W}$ is the vector of concept weights, these concept weights give the importance of a concept for a given class. This means the concept weights are determined on the class level, while concept feature values are different for each image. Additionally $\Omega(g)$ is the (multiclass) elastic-net penalty. Defined as $\Omega(g) = \alpha||\boldsymbol{W}||_1 + (1-\alpha)||\boldsymbol{W}||_2^2$ with $\alpha$ as a regularization factor. Furthermore, if $g$ is a classification problem, we apply softmax to its output. Moreover, $K$ denotes the number of classes in the classification problem and $\lambda$ gives the regularization strength.

There are cases where the concept set is not expressive enough for the PCBMs alone to recover the original model's performance. To solve this, the original authors reintroduce the embeddings to "fit the residuals." They thus introduce Hybrid Post-Hoc CBMs (PCBM-h) to solve the following optimization problem:

$$\min_r \ \mathbb{E}_{(x,y)\sim D}[\mathcal{L}(g(f_C(x)) + r(f(x)), y)], \tag{2}$$

where $r : \mathbb{R}^d \to Y$ represents the residual predictor mapping embeddings to the classes $Y$ in our classification task. Also, minimizing the loss in Equation 2 is a sequential process where both the concept bank and the interpretable predictor $g$ remain fixed, while a linear layer is employed to fit the residuals.

For the claim regarding model editing, Yuksekgonul et al. (2023) again use both PCBM and PCBM-h, with the model editing performed on the sparse linear model $g$. Both the user study and the controlled experiment use the CLIP-ResNet50 backbone model from Radford et al. (2021).

## 3.2 Datasets

In total, the original authors used seven different datasets for experimentation, either to evaluate the performance of PCBMs across different domains, the quality of generated CLIP concepts, or the results of global model editing. All datasets used for binary classification were evaluated using the Area-Under-Curve (AUC), the multi-class binary classification-based COCO-Stuff using the mAP, and the rest using accuracy. An overview of each dataset and its purpose can be found in Table 8.

For COCO-Stuff and SIIM-ISIC, we followed the original paper to create subsets for each to reduce the required disk space for experimentation.[1] The specifications for how they were created can be found in our repository. Meanwhile, for the model editing experiments and the survey, multiple datasets were generated using Metashift with the Visual Genome dataset.[2]

The concepts used for evaluating the performance across different domains were taken from 3 different datasets. For CIFAR-10, CIFAR-100, and COCO-Stuff, the concept set was taken from BRODEN (Fong & Vedaldi, 2018), which contains visual concepts including for objects, settings, and image qualities, among others. Meanwhile, the version of the CUB dataset used from Koh et al. (2020) is annotated with pre-processed bird-related concepts, and the medical concepts for the HAM10000 and SIIM-ISIC originated from the Derm7pt dataset (Kawahara et al., 2019).

For the experiments that evaluated the use of CLIP-generated concepts, ConceptNet was employed to generate the utilized concepts (Speer et al., 2017). As an open knowledge graph, it can be used to find concepts with particular relations for any given query concept, such as "wheel" and "engine" for "car." Similar to the original authors, only five relations sets were used to build the concept space for each class, which are the "hasA," "isA," "partOf," "HasProperty," and "MadeOf" relations.

## 3.3 Hyperparameters

For a comparable replication, we used the same hyperparameters specified in the original paper whenever they were. This was the case for everything apart from the regularization parameters $C_{svm}$ and $\lambda_{regression}^{logistic}$. $C_{svm}$ is used by the SVM for CAV computation. The open-source repository supplies the majority of the necessary code, including an example grid for fine-tuning $C$ values, which is the following: [0.001, 0.01, 0.1, 1.0, 10.0]. Meanwhile, $\lambda_{regression}^{logistic}$ is employed when investigating the original models for CIFAR10, CIFAR100,

---

[1]The trimmed-down datasets can be found here: COCO-Stuff, SIIM-ISIC.
[2]The generated datasets can be found here: Model editing, Survey.

and COCO-Stuff. The original model is CLIP-ResNet50 for these three datasets, thus we determine the hyperparameter in the same way utilized by Radford et al. (2021). As such, we conduct a hyperparameter sweep on validation sets over a range from $10^{-6}$ to $10^6$, with 96 logarithmically spaced steps.

A hyperparameter warranting some attention is the regularization strength $\lambda$. It determines the weight decay for the parameters of our sparse linear model $g$ as defined in subsection 3.1, influencing the interpretability of the model and thus the PCBM. Appendix A of the original paper specifies the values used per dataset, tuned on a validation set, but omits the metric used for tuning. Additionally, the original code notes a trade-off between interpretability and accuracy for this parameter, which is discussed in subsection 4.1.

### 3.4 Experimental setup and code

To test the first two main claims outlined in section 2, we perform reproductions of the original paper's results for CAVs and CLIP concepts using the authors' code repository as well as the same datasets, parameters, backbone models, and number of training epochs used for the PCBMs outlined in their paper. An overview of the backbones and datasets used can be found in Appendix A.

In addition, we test the third claim by evaluating their techniques for doing the model-editing experiments, using an unedited PCBM as a baseline. As such, we replicate their methodology by utilizing 6 scenarios where we artificially introduce a spurious correlation between a class and a concept. For example, in one scenario we use a dataset where all images of a bed contain dogs for training and test the resulting model on a dataset where they contain cats instead. A description of these scenarios can be found in Appendix D.

The first technique we evaluate is called "pruning," where we set the corresponding weights for "bed" and "dog" to zero in the PCBM layer following the above example, ideally resulting in the model assigning less importance to "dog" when predicting "bed." Moreover, we evaluate "pruning with normalization," which is pruning combined with a re-scaling of the weight vector corresponding to the affected class, thus making the new L1 norm match the previous one. We evaluate these methods alongside "fine-tuning," which involves first training a PCBM on the training domain, then fine-tuning in the test domain, and then testing on a held-out set of the test domain.

Another way we evaluate their third claim is through a survey. Here, we use a dataset similar to the one mentioned previously, with the primary distinction being the absence of spurious correlations in the test class. For example, we use images of beds containing dogs for training and then only use bed images without dogs during testing. In addition, we reproduce the survey in a comparable setting to the one presented in the original paper. Using a variant of the base PCBM, we identify the top 10 ranked concept weights for each class of interest and guide the users through the model editing process. Most participants (94.11%) were machine learning students or practitioners and had a median age of 24, alongside a male-to-female ratio of 76.2% to 23.8%. Additionally, informed consent was obtained from all participants.

We follow the original baselines to obtain an accurate benchmark: random pruning (randomly select a subset of the top ten concepts to prune, matching the number of concepts pruned by the user for fair comparison) and greedy pruning (greedily select from the top 10 concepts that improve model accuracy when pruned, again matching the number of concepts pruned by users). Further details of the setup and survey questions can be found in Appendix F.

#### 3.4.1 Assumptions made

Unfortunately, due to missing implementations and details, we had to update the repository to include additional components such as dataset installation scripts, code for the missing implementations, and an environment file. Moreover, we had to make several assumptions in our experimentation:

1. For the COCO-Stuff experiments, binary cross-entropy loss minimization was approached as 20 binary classifications. Performance was averaged across runs, treating the target class as positive and the remaining classes as negative.

2. For the SIIM-ISIC experiments, we implemented our data selection method based on the limited details provided by the authors. These details state that they utilized 2000 images (400 malignant,

1600 benign) for training and 500 images (100 malignant, 400 benign) for model evaluation. on a held-out set of 500 images (100 malignant, 400 benign).

3. For the global model editing experiments, design flexibility existed for the dataset, training set size, class names, optimizers, and hyperparameters due to conflicting information. Examples of this are: $\frac{\lambda}{170 \times 5} = 0.05$ vs $\frac{\lambda}{170 \times 5} = 0.002$, CLIP ResNet50 vs ResNet18, and Adam vs SGD Optimizer. After testing many configurations, we decided to proceed using $\lambda = 1.7$, an L1 ratio of $\alpha = 0.99$, BRODEN concepts, a CLIP-ResNet50 encoder, and the SGD optimizer.

4. For the user study experiments, insights from the Metashift study guided us in selecting hyperparameters and matching the reported weight magnitudes. We experimented with the regularization and settled on $\lambda = 0.002$, which made our results consistent with the original. Also, the architecture, which was built on existing models, now included adjustments and additional implementation of the three pruning methodologies.

## 3.5 Additional experiments

To further assess the efficacy of PCBMs, we explore three research directions. The first investigates PCBMs' interpretability and performance using randomly generated concept vectors. Meanwhile, the second involves the interpretability of the concepts used in PCBMs. Lastly, the third direction aims to verify the original authors' claim that any neural network can be converted into a PCBM by testing it for audio classification.

### 3.5.1 Random projection

Random projections, known for their desirable dimensionality reduction properties, preserve input data distance and similarity (Dasgupta, 2000; Bingham & Mannila, 2001; Cannings, 2021). For this experiment, we substitute the embedding matrix $C$ with a random one, where each $i$'th row is a normally distributed and normalized vector. In other words, we replace the original concept matrix, obtained by the original author's methods, with a random and meaningless concept matrix. The exact details of this random projection experiment are as follows. We train a PCBM and PCBM-h on CIFAR10, CIFAR100, COCO-Stuff, and HAM10000. For the first three datasets, our new random concept matrix has the same dimensionality as the CLIP concept matrix. Meanwhile, for HAM10000, we have eight random concepts, similar to the number of concepts used in the original paper. The results of this experiment will determine whether meaningful concepts are important for only performance, interpretability, or both. Additionally, this experiment gives a baseline on the performance of PCBMs.

### 3.5.2 Object-concept correspondence

We examine whether the concepts used in PCBMs correctly correspond to objects in the input space. To see what this means in practice, we give the following general interpretation of the interpretability of PCBMs. When we refer to the PCBM as interpretable, we implicitly rely on the following two assumptions:

1. The concept weight of concept $i$ for class $k$ is high when $i$ is important for correctly classifying $k$.
2. The concept feature value of concept $i$ for image $j$, reflects **ONLY** the visibility of $i$ in $j$.

The first assumption immediately holds for the used class of interpretable models. However, the second one is less trivial. Assumption two is related to specifically the interpretability of the concepts used. In practice, it means that when we have an image of a bicycle the concept feature value for the concept "green" should only be high when the bicycle is green. Thus issues arise when representations of different concepts become entangled since they will be projected onto the image in a similar way.

Kim et al. (2018) experimentally show that higher-ranked images represent the concept better when ranking images based on the concept feature value. However, they do use a different projection method called the TCAV-score, meaning their findings do not directly carry over to our scenario. Also, Kim et al. (2023) show that CAVs are sensitive to the chosen negative and positive samples, leading to unintended entanglement of concept representations. They find that this happens when the positive images for a concept almost always include a different concept, such as how images of a bicycle frame will almost always include a handlebar.

For CLIP concepts, some potential issues exist relating to assumption two. Firstly, CLIP is trained for single-label classification, whereas we are trying to identify multiple parts of an object in an image. Secondly, a type of entanglement has also been found in CLIP concept representations which involves it treating images as bags of concepts, where any one concept from the bag can be used to explain an entire part of the image (Tang et al., 2023; Lewis et al., 2024). Also, Tang et al. (2023) specifically discovered that there exists a Concept Association Bias (CAB) within CLIP between objects and their attributes, showing that this holds for part-whole relationships and colour. As such, the textual representation of only one of these two can be used to explain the occurrence of both in the image.

Our first method to test object-concept correspondence is by testing assumption two for PCBMs. We do so by evaluating the concept feature values $f_C(\boldsymbol{x})$ for the positive and negative concept images in BRODEN and CUB. We do this for CAVs, CLIP concepts, and random concept vectors, with the latter having the same shape as the CLIP concepts. For each method, we have a set of concepts $I = \{i_1, i_2, \ldots, i_{N_c}\}$, $N_c$ is the number of concepts we obtained using this method. We have two image sets for each concept $P_i = \{\boldsymbol{x_{p_1}}, \ldots, \boldsymbol{x_{p_{50}}}\}$ and $N_i = \{\boldsymbol{x_{n_1}}, \ldots, \boldsymbol{x_{n_{50}}}\}$, each containing 50 positive and 50 negative images for a concept $i$ respectively. We report the following statistic for each method:

$$gap = \frac{1}{|I|} \sum_{i \in I} \left( \frac{1}{50} \sum_{\boldsymbol{pos} \in P_i} f_C(\boldsymbol{pos}) - \frac{1}{50} \sum_{\boldsymbol{neg} \in N_i} f_C(\boldsymbol{neg}) \right). \tag{3}$$

This means that if this assumption holds, we should see generally higher concept feature values for positive concept images than for negative ones, alongside a positive average gap when using the interpretable concept vectors (CAVs and CLIP concepts). Meanwhile, for random concepts, we should not see a difference.

Meanwhile, our second method involves using saliency maps (Simonyan et al., 2014; Smilkov et al., 2017) to see exactly where in the image our model "sees" the concept. We construct saliency maps from the concept projection to the input. Due to this, our saliency maps visualize the gradient of the concept feature value concerning the input. Below we define a saliency map for concept $c$ concerning the input $\boldsymbol{x}$ as $M_c(x)$:

$$M_c(\boldsymbol{x}) = \partial f_c(\boldsymbol{x}) / \partial \boldsymbol{x}. \tag{4}$$

Additionally, we employ the SmoothGrad implementation of saliency maps (Smilkov et al., 2017), which enhances these maps by averaging gradients over multiple forward passes while adding noise to the input $\boldsymbol{x}$. Consequently, our saliency maps depict the sensitivity of concept feature values to input changes, with brighter regions indicating higher importance.

We investigate if the most crucial parts of an image for a given concept visually align with it. For instance, the saliency map for the concept "green" should highlight the green parts of a bicycle in an image. To conduct this experiment, we utilize CIFAR100 images and the CLIP backbone, meaning the model from which we compute gradients comprises the CLIP ResNet50 backbone followed by the concept projection described in subsection 3.1, where the concept matrix $\boldsymbol{C}$ contains the concept vector of a single concept. We generate saliency maps for ten different concepts, with five using CAV concept vectors and the remaining using CLIP ones. For both methods, we select the four concepts with the highest concept weight for the class, along with one hand-chosen concept corresponding to a co-occurring object in the image. This hand-chosen concept aids in examining the less fine-grained behavior of concept vectors.

### 3.5.3 Audio classification

The third experiment aims to verify the original authors' claim that any neural network can be converted into a PCBM. We therefore evaluate its performance in audio classification, a domain that has yet been untested. AudioCLIP was utilized for this purpose, which is an implementation of CLIP that has been integrated with a compatible and well-performing audio encoder (Guzhov et al., 2022). This encoder is an ESResNeXt instance that has been jointly trained alongside the other CLIP encoders (Guzhov et al., 2021).

We assess the audio model performance on ESC-50 (Salamon et al., 2014) and UrbanSound8K (Piczak, 2015). Similar to the models used in the main reproduction study, we create a baseline for AudioCLIP's performance via linear probing. To obtain the CLIP-based concept matrix, we collect textual concepts and feed them through the AudioCLIP text encoder, similar to how it was done with standard CLIP.

We derive textual concepts as follows: From ESC-50, ConceptNet generates 146 concepts from ground-truth labels, following the original authors' method. UrbanSound8K provides 31 concepts from non-terminal nodes with depths 1 to 4 in its taxonomy, excluding terminal nodes with data point labels. The final set of concepts is derived from the AudioSet ontology, totalling 610 concepts after excluding class-label-containing nodes. Thus, the concept matrix has a total of 787 concept vectors.

To obtain the CAV-based concept matrix, we perform the following weakly supervised approach: For each label in a dataset, we generated its associated concepts using ConceptNet. Then, we gathered 50 audio clips for each concept whose labels were associated with the given concept and treated them as positive examples, using a similar process for negative examples. Afterwards, we proceeded to train a SVM to separate positive and negative examples and took the vector normal to the linear boundary as the concept's vector. This approach lead to 31 concepts for UrbanSound8K and 171 for ESC-50.

We evaluate PCBM and PCBM-h on various datasets using CAV and CLIP-based concept matrices. The AudioCLIP audio head's final embedding dimension has a size of 2048, making the residual classifier of PCBM-h a function $r : \mathbb{R}^{2048} \to N_{classes}$. For UrbanSound8K, we set $\lambda_{CLIP} = 2 \cdot 10^{-4}$ and $\lambda_{CAV} = \frac{1}{34 \times 10}$. Similarly, for ESC-50, $\lambda_{CLIP} = 2 \cdot 10^{-10}$ and $\lambda_{CAV} = \frac{1}{171 \times 50}$. Early stopping was applied to the ESC-50 models due to observed overfitting.

### 3.6 Computational requirements

All of our experiments were conducted using Google Colab in region "europe-west4," which has a carbon efficiency of 0.57 kgCO$_2$eq/kWh. However, most experiments were CPU-based as almost all training and evaluation was done only to evaluate the PCBM performances given the usage of pre-trained models for the backbones. As such, only the PCBM-h instances required GPU computation as they are neural networks. We utilized a T4 GPU and Intel(R) Xeon(R) CPU for these experiments, resulting in a total computational cost of roughly 30 CPU and 30 GPU hours for all experiments. This would amount to 2.39 kgCO$_2$eq in emissions from GPU usage which was entirely offset by the cloud provider. These estimates were conducted using the Machine Learning Impact calculator presented in Lacoste et al. (2019).

## 4 Results

### 4.1 Results reproducing original paper

**Claim 1** - One of the main claims originally made is that PCBMs achieve comparable or sometimes even identical performance to the original baselines across various models. Our results (shown in Table 1) mostly support this claim, as we observed results similar to the original for all datasets besides CUB, which returned a performance that was 10 percentage points higher, aligning more with the CUB ResNet18 pre-trained model's performance.[3] This suggests a potential mistake in the author's evaluation code. In any case, our results show that the performances achieved and trends found match those of the original study quite closely.

| Original | CIFAR10 | CIFAR100 | COCO-Stuff | CUB | HAM10000 | SIIM-ISIC |
|---|---|---|---|---|---|---|
| Original Model | 0.888 | 0.701 | 0.770 | 0.612 | 0.963 | 0.821 |
| PCBM | $0.777 \pm 0.003$ | $0.520 \pm 0.005$ | $0.741 \pm 0.002$ | $0.588 \pm 0.008$ | $0.947 \pm 0.001$ | $0.736 \pm 0.012$ |
| PCBM-h | $0.871 \pm 0.001$ | $0.680 \pm 0.001$ | $0.768 \pm 0.01$ | $0.610 \pm 0.01$ | $0.962 \pm 0.002$ | $0.801 \pm 0.056$ |

| Reproduction | CIFAR10 | CIFAR100 | COCO-Stuff | CUB | HAM10000 | SIIM-ISIC |
|---|---|---|---|---|---|---|
| Original Model | 0.885 | 0.699 | 0.838 | 0.744 | 0.963 | 0.761 |
| PCBM | $0.773 \pm 0.001$ | $0.511 \pm 0.002$ | $0.796 \pm 0.005$ | $0.577 \pm 0.007$ | $0.926 \pm 0.004$ | $0.511 \pm 0.002$ |
| PCBM-h | $0.883 \pm 0.002$ | $0.688 \pm 0.002$ | $0.797 \pm 0.001$ | $0.595 \pm 0.005$ | $0.957 \pm 0.004$ | $0.751 \pm 0.021$ |

Table 1: Original (Yuksekgonul et al., 2023) and reproduction results of the claim that PCBMs achieve comparable performance to the original model. The reported metrics are AUROC for HAM10000 and SIIM-ISIC, mAP for COCO-Stuff, and accuracy for CIFAR and CUB.

---

[3]The pre-trained model and its performance can be found here: https://github.com/osmr/imgclsmob

**Claim 2** - We evaluate if PCBMs could achieve at least the same, if not higher, performance using generated concepts. However, our findings (as shown in Table 2) contradict this claim, evidenced by how for both CIFAR10 and CIFAR100, the PCBM results are approximately 10 percentage points lower than originally reported. Notably, the PCBM for CIFAR10 performed worse when using CLIP concepts compared to BRODEN concepts, suggesting potential limitations in the expressiveness of CLIP concepts. Moreover, our results indicate that the performance reported in the original paper could be replicated using a lower $\lambda$ value than mentioned. This is further explained in Appendix B.

It is noteworthy that the PCBM-h with CLIP concepts achieves comparable performance to CAVs, which is expected due to the sequential procedure in addition to the reintroduction of the embeddings when training PCBM-h. Because of this, the performance of PCBM-h depends only minimally on the performance of PCBM, and by extension, the concepts used. Furthermore, Figure 3 (in Appendix C) displays example concept weights for the same classes and datasets as in the original paper. While the HAM10000 concept weights are quite similar, the weight values for CIFAR100 are one order of magnitude higher (though the important weights are similar). This aligns with the potential use of a lower $\lambda$, as lower weight decay typically results in larger concept weights.

| Original | CIFAR10 | CIFAR100 | COCO-Stuff |
|---|---|---|---|
| Original Model | 0.888 | 0.701 | 0.770 |
| PCBM & labeled concepts | $0.777 \pm 0.003$ | $0.520 \pm 0.005$ | $0.741 \pm 0.002$ |
| PCBM-h & labeled concepts | $0.871 \pm 0.001$ | $0.680 \pm 0.001$ | $0.768 \pm 0.01$ |
| PCBM & CLIP concepts | $0.833 \pm 0.003$ | $0.600 \pm 0.003$ | $0.755 \pm 0.001$ |
| PCBM-h & CLIP concepts | $0.874 \pm 0.001$ | $0.691 \pm 0.006$ | $0.769 \pm 0.001$ |

| Reproduction | CIFAR10 | CIFAR100 | COCO-Stuff |
|---|---|---|---|
| Original Model | 0.885 | 0.699 | 0.838 |
| PCBM & labeled concepts | $0.773 \pm 0.001$ | $0.511 \pm 0.002$ | $0.796 \pm 0.005$ |
| PCBM-h & labeled concepts | $0.883 \pm 0.002$ | $0.688 \pm 0.002$ | $0.797 \pm 0.001$ |
| PCBM & CLIP concepts | $0.736 \pm 0.005$ | $0.535 \pm 0.003$ | $0.801 \pm 0.002$ |
| PCBM-h & CLIP concepts | $0.881 \pm 0.002$ | $0.688 \pm 0.004$ | $0.797 \pm 0.001$ |

Table 2: Original (Yuksekgonul et al., 2023) and reproduction results of the claim that PCBMs do not require labeled concept datasets, alongside comparisons to the original performances.

| Original | Unedited | Prune | Prune + Normalize | Fine-tune (Oracle) |
|---|---|---|---|---|
| PCBM Accuracy | $0.656 \pm 0.025$ | $0.686 \pm 0.026$ | $0.750 \pm 0.019$ | $0.859 \pm 0.028$ |
| PCBM Edit Gain | - | 0.029 | 0.093 | 0.202 |

| Reproduction | Unedited | Prune | Prune + Normalize | Fine-tune (Oracle) |
|---|---|---|---|---|
| PCBM Accuracy | $0.864 \pm 0.033$ | $0.874 \pm 0.025$ | $0.873 \pm 0.025$ | $0.939 \pm 0.004$ |
| PCBM Edit Gain | - | $0.010 \pm 0.009$ | $0.009 \pm 0.008$ | $0.075 \pm 0.031$ |

Table 3: Original (Yuksekgonul et al., 2023) and reproduction results of the claim that PCBMs allow for global model editing.

**Claim 3** - While we see that global model editing (both pruning and pruning with normalization) results in improvements relative to the baseline, the performance increase is half that of the original results, as seen in Table 3. Note that the results are averaged over 10 seeds, such that the PCBM edit gain is the average gain across 10 seeds. For a more detailed description of the per-class edit gain, refer to Appendix E. With regards to the user study, we find that users demonstrated enhancements in 4 of the 9 scenarios, as illustrated in our user study (see Table 15 replicating the original paper's Table 9). However, the extent of these improvements does not align with the initial claim (comparing with the original paper's Table 4), as the performance increases are generally modest. In fact, we observed declines in performance for most scenarios, especially when the number of pruned concepts increased (see Table 4).

| Original | Layer | Unedited | Random Prune | User Prune | Greedy Prune (Oracle) |
|---|---|---|---|---|---|
| | PCBM Accuracy | 0.620 ± 0.035 | 0.604 ± 0.039 | 0.719 ± 0.042 | 0.740 ± 0.041 |
| Spurious | PCBM Edit Gain | - | -0.016 | 0.099 | 0.120 |
| Class | PCBM-h Accuracy | 0.642 ± 0.034 | 0.622 ± 0.037 | 0.736 ± 0.034 | 0.766 ± 0.034 |
| | PCBM-h Edit Gain | - | 0.020 | 0.094 | 0.124 |

| Reprod. | Layer | Unedited | Random Prune | User Prune | Greedy Prune (Oracle) |
|---|---|---|---|---|---|
| | PCBM Accuracy | 0.731 ± 0.172 | 0.217 ± 0.270 | 0.199 ± 0.249 | 0.276 ± 0.291 |
| Spurious | PCBM Edit Gain | - | -0.514 | -0.532 | -0.455 |
| Class | PCBM-h Accuracy | 0.740 ± 0.145 | 0.736 ± 0.144 | 0.735 ± 0.144 | 0.733 ± 0.145 |
| | PCBM-h Edit Gain | - | -0.004 | -0.005 | -0.007 |
| | PCBM Accuracy | 0.822 ± 0.038 | 0.737 ± 0.059 | 0.733 ± 0.577 | 0.747 ± 0.066 |
| All | PCBM Edit Gain | - | -0.085 | -0.089 | -0.075 |
| Classes | PCBM-h Accuracy | 0.854 ± 0.056 | 0.857 ± 0.052 | 0.856 ± 0.052 | 0.855 ± 0.053 |
| | PCBM-h Edit Gain | - | 0.003 | 0.002 | 0.001 |

Table 4: Original (Yuksekgonul et al., 2023) and reproduction results of the claim that human-guided editing improves accuracy.

In our study, users pruned 3.11 ± 2.12 concepts on average, showing larger variation compared to the original average of 3.65 ± 0.39. This variance affected both the greedy and random methodologies, as they rely on the number of pruned concepts. Further details on how these techniques were employed and design decisions regarding the initial selection of classes in the multimodal approach can be found in Appendix F and Appendix H respectively. Unfortunately, we could not provide a detailed breakdown of the number of scenarios improved per user, as the observed improvements were mostly minor. Moreover, some scenarios that showed enhancements in one instance also demonstrated declines in performance across many runs.

Nonetheless, the initial claim that users could complete the pruning task faster than the Greedy approach with only a relatively minor decrease in performance indeed holds. Users achieved comparable outcomes (a slight decrease in PCBM score and an increase in PCBM-h) while maintaining a mean time of 47.48 ± 31 seconds per scenario, in contrast to the Greedy approach's average time of 52.49 ± 3.58.

### 4.2 Results beyond original paper

#### 4.2.1 Random projection

The experiment results of using random projections to assess the necessity of meaningful concepts (e.g., CLIP and CAV concepts) for performance and interpretability are displayed in Table 2. PCBM retains substantial performance when using random concepts for CIFAR10, CIFAR100, and COCO-Stuff, indicating the preservation of backbone embeddings information post-projection. However, the HAM10000 performance is not retained, possibly because a random projection with eight concepts cannot maintain enough information about the original dataset.

| | CIFAR10 | CIFAR100 | COCO-stuff | HAM10000 |
|---|---|---|---|---|
| Original Model | 0.885 | 0.699 | 0.838 | 0.963 |
| PCBM | 0.746 ± 0.002 | 0.490 ± 0.003 | 0.810 ± 0.002 | 0.500 ± 0.000 |
| PCBM-h | 0.882 ± 0.002 | 0.685 ± 0.004 | 0.797 ± 0.001 | 0.961 ± 0.004 |

Table 5: Extension results of the random projection experiment.

The concept weights for two classes in CIFAR100 and HAM10000 are shown in Figure 4 and Figure 5 (Appendix C). For CIFAR100, the concept weights for random concepts lack intuitiveness. Meanwhile, all concept weights are zero for HAM10000. This is consistent with the earlier finding that PCBMs fail to

maintain the original model's performance on this dataset, suggesting the necessity of collecting meaningful concepts for straightforward interpretability. Comparing the performance results with those obtained using meaningful concepts, we find that the latter does outperform this baseline in all cases.

### 4.2.2 Object-concept correspondence

This section contains our findings from testing the assumption that a concept feature value reflects the visibility of a concept in the image. The results for this can be found in Table 2, which presents the average concept feature values for BRODEN and CUB. Looking at the results, CAVs seem to have the greatest separation between positive and negative images, with gaps of 0.274 and 1.694 respectively. Meanwhile, the CLIP concept vectors return a notably weaker signal for a concept's appearance, with gaps of 0.053 and 0.022 for BRODEN and CUB respectively. It is especially weak for the latter since we see that there are incorrect and negative signals within one standard deviation.

We also see that random concept vectors have an average of around zero in both cases as expected, given that no relevant information about the images is encoded. Thus, we find that CAV and CLIP concept feature values do indeed represent a signal for whether a concept is in the image.

|  | **BRODEN** | **CUB** |
|---|---|---|
| Concept Activation Vectors | $0.274 \pm 0.124$ | $1.694 \pm 0.037$ |
| CLIP Concept Vectors | $0.053 \pm 0.046$ | $0.022 \pm 0.037$ |
| Random Concept Vectors | $0.002 \pm 0.018$ | $-0.006 \pm 0.088$ |

Table 6: Extension results of the concept feature value experiment. Reported here are the mean and standard deviations of the concept feature value gaps between 50 positive and negative images, averaged across concepts.

Both Figure 1 and Appendix I depict the outcomes of our experiment, revealing the most relevant parts of the image for each concept via saliency maps for two CIFAR100 images. The former illustrates saliency maps for the four most significant concepts of the class "bicycle," along with maps for the concept "grass," chosen due to its co-occurrence with bicycles in the image corners. Notably, most saliency maps mainly emphasize the primary object, regardless of the concept's referenced part of the image. For example, both "chain wheel" and "handlebar" saliency maps for CAVs are bicycle-shaped, consistent with the findings of Kim et al. (2023), which highlight the unintended entanglement in CAVs. In addition, the bicycle-shaped saliency map for the concept "book" likely stems from entanglement, as it is a top 4 concept for the bicycle class despite its apparent irrelevance. Notably, CAV saliency maps exhibit greater diversity compared to CLIP, highlighting different, incorrect parts of the bicycle, possibly due to CAV's sensitivity to sampled positive and negative images, as discussed in Kim et al. (2023).

A similar entanglement is present for the CLIP concepts, as the saliency maps for "bicycle wheel," "coaster brake," "two wheels," and "bicycle seat" are all bicycle-shaped. This matches previous findings showing that CLIP suffers from CAB and thus cannot distinguish well between an object and its attributes (Tang et al., 2023). A related possible reason why our saliency maps do not correctly highlight the regions of the image corresponding to the concepts is that the clip was not trained for such fine-grained classification. A related finding by Zhong et al. (2021) is that CLIP does not transfer well to region detection tasks. They hypothesize that this is because CLIP was trained to only match whole images to the corresponding text descriptions without capturing the fine-grained alignment between image regions and text spans.

Further evidence that our saliency map findings are due to entanglement rather than other factors (i.e., our CLIP-ResNet50 backbone encoder only extracting features related to the main object in a scene) is provided by Chen et al. (2023), who discovered that CLIP-ResNet50 channels exhibit more noise compared to a ResNet50 trained solely on ImageNet. Using various saliency map types, they demonstrated that this backbone examines more than just the main object within a scene. Despite this, a positive observation for PCBMs is that concepts such as "green" and "greenness" successfully highlight the top right green patch of the image, indicating that concept feature values sometimes reflect broader relevant parts of the image.

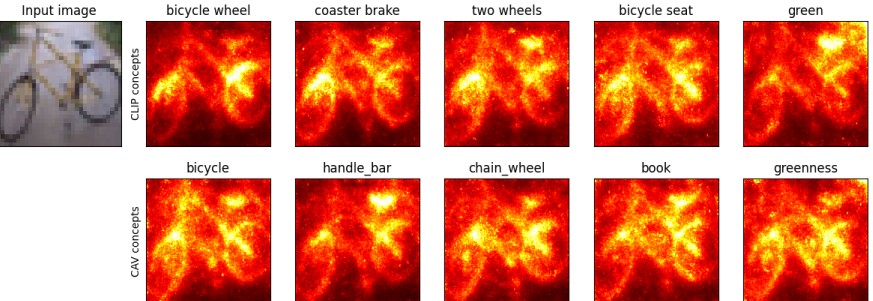

Figure 1: Extension results of the saliency map experiment. Shown here are ten saliency maps for an image from the class "bicycle" of CIFAR100. Five CAV and five CLIP concepts were used.

### 4.2.3   Audio classification

An overview of the audio classification extension results is provided in Table 7. PCBM struggles to match the original model's performance when using UrbanSound8K with CLIP concepts, though PCBM-h narrows this gap. This pattern does not hold, however, for other dataset-method combinations. For example, PCBM-h fails to reach the original model's performance when using UrbanSound8K with CAVs. Meanwhile, CLIP performs poorly for ESC-50, and the CAV-based approach falls short of the original model despite improving over CLIP. The discrepancy in CLIP-based performances may stem from the number of CLIP concepts not scaling with the number of classes in ESC-50, which is something discussed by Oikarinen et al. (2023). This means that PCBMs are harder to scale to datasets with many labels.

The inadequacy of CLIP's concept space is evident in Table 12 (Appendix D), where top concepts like "heavy metal," "motorcycle," and "whirr" are mismatched with their class label "cow." Conversely, CAVs' consistency across datasets may arise from the customized concepts tailored to each dataset. The results in Table 11 and Table 12 (in Appendix D) support the interpretability of CAVs over CLIP. Each class has multiple fitting concepts within the top three most important concepts. Thus, based on this, extending PCBMs to different domains can yield mixed results, heavily influenced by the chosen concept subspace.

|  | **UrbanSound8K** | **ESC-50** |
|---|---|---|
| Original Model | 0.613 | 0.670 |
| PCBM & CLIP concepts | $0.558 \pm 0.002$ | $0.280 \pm 0.035$ |
| PCBM-h & CLIP concepts | $0.603 \pm 0.006$ | $0.280 \pm 0.035$ |
| PCBM & labeled concepts | $0.411 \pm 0.001$ | $0.400 \pm 0.006$ |
| PCBM-h & labeled concepts | $0.462 \pm 0.009$ | $0.410 \pm 0.006$ |

Table 7: Extension results of the audio classification experiment.

## 5   Discussion

Our study aimed to replicate the findings from Yuksekgonul et al. (2023) and somewhat successfully reproduced their first two claims, though had challenges with doing so for the third. The first claim, asserting that Post-Hoc Concept Bottlenecks do not reduce model performance, aligns with our results. Our PCBM experiments show performance similar to that in the original paper and PCBM-h retrieves a similar performance to the baseline, with negligible differences observed in certain scenarios. As outlined before, some of these differences are caused by potential implementation mistakes by the original authors, relating to the conflicting results obtained for CUB. Furthermore, other differences stem from missing details regarding the experiment setup, which is the case for the experiments related to COCO-Stuff and, to an extent, SIIM-ISIC.

The second claim, suggesting the usability and potential advisability of CLIP concepts, is partially supported. While the PCBM-h here performs similarly to the original model, the PCBM results are mixed and not consistently superior to the CAV-based approach. Meanwhile, the third claim regarding PCBMs enabling global model editing is not truly supported by our results. From the Metashift experiment, the observed increase was significantly less than the original results, and the user study indicated only marginal performance improvements and, in many cases, declines. Similar to some of the other experiments performed, this discrepancy is due to incomplete implementation details provided by the authors, especially relating to how the Metashift PCBMs were trained.

Beyond these three claims, we evaluated the interpretability of PCBMs by first testing whether meaningful concepts are necessary for both performance and interpretability, discovering that random, meaningless, concepts perform well but are not easily interpretable. We then examined if the concepts used by the model correctly correspond to objects in the input image using saliency maps and observed that the concept projection generally signals an object's present within an image correctly, though this signal is not based on only the image part containing said object. Based on this, our conclusion is that the interpretability of PCBMs is limited due to how the concept feature values used by the interpretable model are not obtained only by looking at the concept in the image. Instead, many concepts return high feature values only because they are correlated with the main object we are classifying.

Furthermore, another of our extensions partially supports the claim that PCBMs apply to any neural network. We find that results when applying PCBMs to audio are mixed. The CAV-based approach is consistent and gives interpretable concepts, but does not perform as well as the original model. The CLIP-based approach does as well as the original model for one dataset but does poorly on the other. Additionally, the CLIP concepts are harder to interpret in combination with the classes.

Given these results, we recommend further research focusing on the interpretability of both concept vectors discussed. Additionally, exploring more suitable projection methods for CLIP concepts is crucial to enhance their responsiveness in the presence of specific concepts in images. Another recommendation is to evaluate the performance of additional audio datasets and architectures to investigate if other, non-CLIP-based audio classifiers still function with the PCBM architecture. Lastly, given how frequently PCBMs are utilized (Oikarinen et al. (2023); Yang et al. (2023); Panousis et al. (2023); Daneshjou et al. (2022b)), we suggest that future researchers be more careful when using them, as our findings show that proper interpretability is not fully guaranteed. It would be especially beneficial to examine further research utilizing PCBMs for object attribute entanglement in concepts, to see if they suffer from the same issues covered here.

### 5.1 What was easy, and what was difficult?

An appendix with all used hyperparameters and implementation details was provided by the original paper. This made it simple to start with the experiments, especially when coupled with the publicly available repository and the complete model implementations.

That aside, the repository provided was still incomplete, meaning that several code parts, such as the COCO-Stuff implementations, model editing files, and Metashift concept sources, were unavailable. The explanation for some of the experiment setups is also insufficient for producing accurate reproductions, resulting in the assumptions outlined in subsection 3.4.

### 5.2 Communication with original authors

We have attempted to contact the original authors for clarification on how some of the experiments were set up due to missing details/implementations, such as how the COCO-Stuff binary classification and Metashift experiments were performed. However, as of writing, they have not responded to any of our inquiries.

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

## A  Specifications of datasets used

Table 8 contains an overview of all datasets used by the original authors, which were experimented on for reproduction.

| Task | Dataset | Backbone model | Concepts used | Number of concepts used | Classes |
|------|---------|----------------|---------------|------------------------|---------|
| Evaluating performance across different domains | CIFAR-10 CIFAR-100 COCO-Stuff | CLIP-ResNet50 | BRODEN (Fong & Vedaldi, 2018) | 170 | 10 100 20 |
| | CUB | ResNet18 | From Koh et al. (2020) | 112 | 200 |
| | HAM10000 SIIM-ISIC | Inception | Derm7pt (Kawahara et al., 2019) | 8 | 2 |
| Evaluating generated concepts | CIFAR-10 CIFAR-100 COCO-Stuff | CLIP-ResNet50 | ConceptNet generated concepts | - - - | 10 100 20 |
| Controlled Metashift Experiments For Model Editing | Metashift | CLIP-ResNet50 | BRODEN (Fong & Vedaldi, 2018) | 170 | 5 |
| User Study | Metashift | CLIP-ResNet50 | ConceptNet generated concepts | 440 | 5 |

Table 8: Overview of datasets used from the original study.

As part of the audio classification extension, the ESC-50, UrbanSound8K, and AudioSet datasets were utilized (Salamon et al., 2014; Piczak, 2015). All datasets were created to tackle the issue of data scarcity in automatic urban sound classification and are used to further fine-tune the AudioCLIP audio encoder (Guzhov et al., 2022). An overview of them can be found in Table 9.

| Dataset | Backbone model | Concepts used | Number of concepts used |
|---------|----------------|---------------|------------------------|
| ESC-50 | | Generated by ConceptNet | 146 |
| UrbanSound8K | AudioCLIP | From Salamon et al. (2014) | 31 |
| AudioSet | | From Gemmeke et al. (2017) | 610 |

Table 9: Overview of the datasets used for audio classification.

# B  Accuracy/Interpretability trade-off

Table 10 shows the different PCBM accuracies when varying the regularization strength. We see there that for both CIFAR10 and CIFAR100, the accuracy is highest when the regularization strength is $\frac{0.1}{KN_c}$, after which it decreases. To examine the trade-off between accuracy and interpretability that exists when tuning the regularization strength, we plot the regularization strength against the number of non-zero parameters and the absolute sum of weights. For both metrics lower is better. Figure 2a and Figure 2b show these plots for CIFAR10, where we see that a lower regularization strength leads to the trained PCBM doing worse on the interpretability metrics. As such, the regularization strength can not be decreased to obtain higher accuracies without harming interpretability. Figure 2c and Figure 2d show similar trends for CIFAR100.

| Regularization strength | $\frac{10.0}{KN_c}$ | $\frac{1.0}{KN_c}$ | $\frac{0.1}{KN_c}$ | $\frac{0.01}{KN_c}$ | $\frac{0.001}{KN_c}$ |
|---|---|---|---|---|---|
| CIFAR10 | 0.318 | 0.552 | 0.559 | 0.507 | 0.461 |
| CIFAR100 | 0.648 | 0.793 | 0.839 | 0.824 | 0.820 |

Table 10: PCBM accuracies for different regularization strengths.

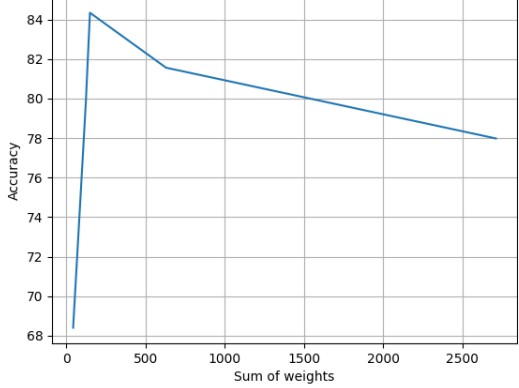

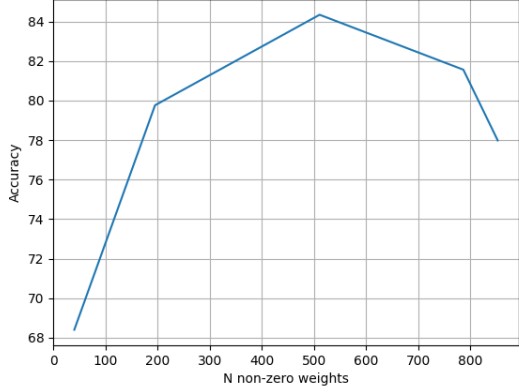

(a) The absolute sum of all class weights against accuracy for a PCBM trained on CIFAR10.

(b) The number of non-zero weights against accuracy for a PCBM trained on CIFAR10.

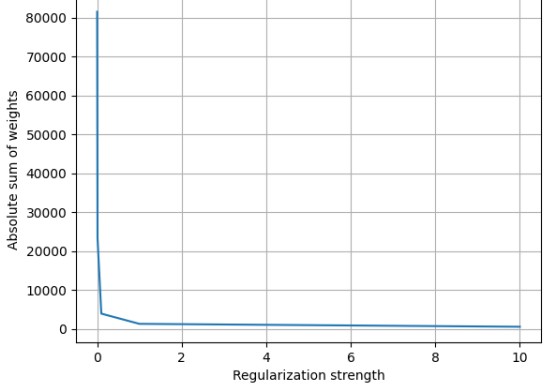

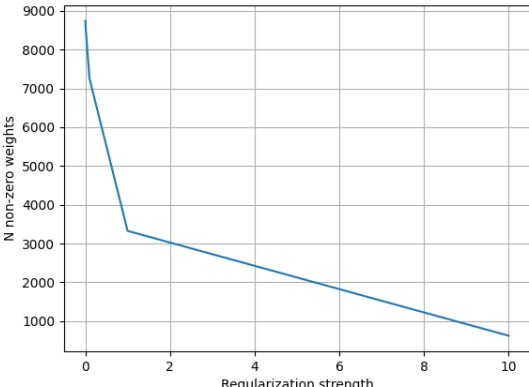

(c) The absolute sum of all class weights against accuracy for a PCBM trained on CIFAR100.

(d) The number of non-zero weights against Accuracy for a PCBM trained on CIFAR10.

Figure 2: Tradeoffs between accuracy, class weights, and regularization strength for selected scenarios.

## C    Example weights obtained

Below are examples of weights obtained for certain classes from the HAM10000, CIFAR100, and audio experiments. As mentioned in the section 4, we see that the obtained concept weights are indeed human-logical, which supports the usability of PCBMs for interpretability.

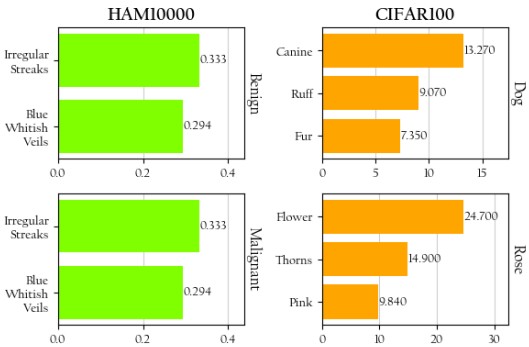

Figure 3: Example weights for classes of HAM10000 (CAVs) and CIFAR100 (CLIP concepts).

We also attach examples of weights obtained from some of the random projection experiments and for CIFAR100. Both figures can be found below:

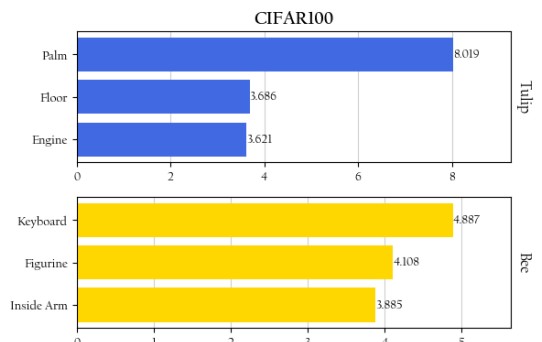

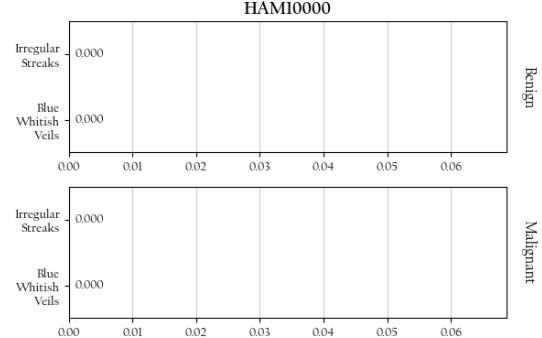

Figure 4: Example weights for classes of CIFAR100 (CLIP concepts).

Figure 5: Example weights for classes of HAM10000 (Random Concept Vectors).

## D    Preprocessing Metashift for model editing experiments

To generate the 10 scenarios used in the model editing experiments, the original authors used Metashift. They first defined two 5-way classification tasks where they specified different independent scenarios for each. Furthermore, every scenario contained one class which is spuriously correlated with a different concept in its train and test datasets, and each split consists of 100 images per class. For instance, the first scenario of the first task in Table 13 consists of a training set where all images corresponding to the class bed contain a dog and a test set where all images of beds contain cats along with 100 unconstrained images for each remaining class.

We can see in Table 13 each task with its 5 classes and corresponding scenarios. Although there is conflicting information on the number of images per class (either 50 or 100), it is worth noting that of the 10 scenarios present, four of them do not have 100 images. We considered that 250 images per split was too little, so we discarded the four problematic scenarios.

| Class | CLIP Concepts (weights) | Labeled Concepts |
|---|---|---|
| air_conditioner | Doorbell (17.85) | machine (10.698) |
| | Bicycle bell (17.81) | automobile (10.238) |
| | Cat communication (16.87) | air (4.116) |
| car_horn | Shofar (14.49) | honk (4.858) |
| | Sonar (12.40) | bullet (0.334) |
| | Harmonic (11.56) | bang (0.297) |
| children_playing | Babbling (20.03) | toy (6.711) |
| | Hubbub, speech noise, speech babble (19.47) | school (5.524) |
| | Chatter (18.26) | fun (5.484) |
| dog_bark | Growling (29.54) | cry (12.929) |
| | Yip (27.51) | dog (10.003) |
| | Bow-wow (21.07) | woof (8.170) |
| drilling | Cattle, bovinae (22.48) | construction (4.338) |
| | Honk (17.71) | machine (0.000) |
| | Canidae, wolves (13.47) | air (0.000) |
| engine_idling | Purr (19.06) | automobile (7.059) |
| | Meow (17.67) | machine (6.176) |
| | Claws (16.95) | motorcycle (5.037) |
| gun_shot | Silence (14.86) | bang (4.549) |
| | Arrow (13.78) | bullet (4.545) |
| | Chainsaw (12.55) | honk (1.769) |
| jackhammer | Snake (23.94) | rock (5.604) |
| | Steelpan (19.34) | drill (4.969) |
| | Tambourine (18.31) | construction (0.166) |
| siren | Ice cream truck, ice cream van (28.41) | wail (16.096) |
| | Air horn, truck horn (27.81) | alarm (9.479) |
| | Ambulance (27.61) | noise (5.034) |
| street_music | Shofar (16.56) | entertainment (8.744) |
| | Water tap, faucet (16.43) | automobile (4.411) |
| | Clarinet (14.77) | rap (3.860) |

Table 11: Comparison of the top three concept weights between CAVs (labeled concepts) and concepts obtained using CLIP for UrbanSound8K.

Furthermore, it is unclear to us how the domain cow(cat) could even be generated, as the Metashift domain lookup table[4] does not register any such image and has not been changed since the creation of the Metashift GitHub repository. We speculate that the original authors may have used a different base dataset (such as COCO) for Metashift, but we chose to stick with the Visual Genome dataset as it was the default option and because no custom base datasets were mentioned in the original paper.

We make the chosen six scenarios available on HuggingFace along with the entirety of our deterministic pre-processing pipeline for full transparency. Although we report all of our results using a cherry-picked dataset (meaning that we manually pick the best images that represent the domain shift), we also provide a randomly selected dataset.

---

[4]`https://github.com/Weixin-Liang/MetaShift/blob/main/dataset/meta_data/full-candidate-subsets.pkl`

| Class | CLIP Concepts (weights) | Labeled Concepts |
|---|---|---|
| dog | Growling (3260.187) | canine (9.666) |
| | Yip (2679.440) | woof (8.258) |
| | Bow-wow (2495.401) | four_legged_animal (7.040) |
| rooster | Power windows, electric windows (2253.037) | crowing (12.931) |
| | male chicken (1976.757) | chicken (10.502) |
| | Crowing, cock-a-doodle-doo (1950.169) | farm_animal (10.376) |
| pig | Ping (2314.484) | engine (4.299) |
| | Chewing, mastication (2257.323) | amphibian (3.932) |
| | Snort (2160.123) | oink_oink (2.580) |
| cow | Heavy metal (2201.455) | moo (12.866) |
| | Motorcycle (2009.351) | living_creature (10.481) |
| | Whir (1978.380) | nature (7.279) |
| frog | Pig (2828.164) | amphibian (18.596) |
| | Ringtone (2783.966) | ribbit (17.952) |
| | Frog (2761.661) | windshield (4.473) |
| cat | Cat (2598.395) | feline (10.581) |
| | Cat communication (2558.293) | meow (8.726) |
| | Caterwaul (2385.926) | scratch_furniture (7.948) |
| hen | Pigeon, dove (1894.772) | chirps (7.573) |
| | Dishes, pots, and pans (1873.768) | bird (4.638) |
| | Chink, clink (1797.078) | farm_animal (3.759) |
| insects | Strum (1400.697) | move (7.170) |
| | Whale vocalization (1386.632) | bugs (6.840) |
| | Croak (1372.076) | animals (4.380) |

Table 12: Comparison of the top three concept weights between CAVs (labeled concepts) and concepts obtained using CLIP for ESC-50.

| Task | Train domain | Test domain | Availability |
|---|---|---|---|
| | bed(dog) | bed(cat) | Enough images |
| **Task 1** | bed(cat) | bed(dog) | Enough images |
| (airplane, bed, | car(dog) | car(cat) | $50 < \text{car(cat)} < 100$ |
| car, cow, | car(cat) | car(dog) | $50 < \text{car(cat)} < 100$ |
| keyboard) | cow(dog) | cow(cat) | $\text{cow(cat)} = 0$ |
| | keyboard(dog) | keyboard(cat) | $0 < \text{keyboard(dog)} < 50$ |
| **Task 2** | table(cat) | table(dog) | Enough images |
| (beach, computer, | table(dog) | table(cat) | Enough images |
| motorcycle, | table(books) | table(dog) | Enough images |
| stove, table) | table(books) | table(cat) | Enough images |

Table 13: Metashift task splits and their availability.

# E    Per scenario performance on model editing experiments

Similar to Table 6 of the original paper, we compute the accuracies and standard error of each scenario and method as described in Appendix D. As shown in Table 14, both pruning and pruning with normalization do not show the improvements seen in the original study, even when averaging over 10 seeds and trying all the ambiguous configurations of regularization strengths and backbones mentioned. (more information on Figures 6a and 6b). For Table 14 we use CLIP's pre-trained ResNet50 with a $\lambda$ of 1.7 (which results in an overall regularization strength of 0.002), learning rate of 0.5, $\alpha$ of 0.99 and using the SGDClassifier.

In Figure 6, we see that no matter the regularization strength or the backbone used, the accuracy change does not compare to the original paper. In addition, the ResNet18 accuracy change is negligible at best.

| Train | Test | Original | Prune | Prune + Normalize | Finetune |
|-------|------|----------|-------|-------------------|----------|
| bed(cat) | bed(dog) | 0.928 ± 0.001 | 0.930 ± 0.001 | 0.930 ± 0.001 | 0.946 ± 0.001 |
| bed(dog) | bed(cat) | 0.926 ± 0.001 | 0.925 ± 0.001 | 0.925 ± 0.001 | 0.943 ± 0.001 |
| table(books) | table(cat) | 0.761 ± 0.002 | 0.795 ± 0.004 | 0.792 ± 0.003 | 0.924 ± 0.001 |
| table(books) | table(dog) | 0.765 ± 0.002 | 0.805 ± 0.002 | 0.804 ± 0.002 | 0.948 ± 0.001 |
| table(cat) | table(dog) | 0.923 ± 0.002 | 0.918 ± 0.003 | 0.918 ± 0.003 | 0.946 ± 0.001 |
| table(dog) | table(cat) | 0.883 ± 0.001 | 0.874 ± 0.001 | 0.872 ± 0.002 | 0.927 ± 0.002 |

Table 14: Accuracy and Standard Error for Metashift model editing tasks over 10 seeds.

To make sure that our implementation does not differ substantially from the intended, we make two more: First, one that is as close as possible to the existing author's code, which we'll name Strict ResNet50, and second, an implementation that uses the Adam Optimizer for the training procedure (as said in the original paper), which we'll name Adam ResNet50. In Figures 7a and 7b we can observe that both implementations perform considerably worse than ours.

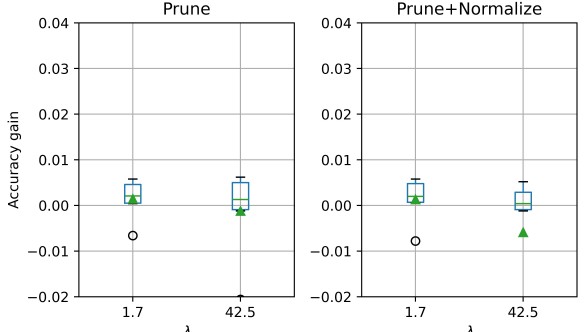
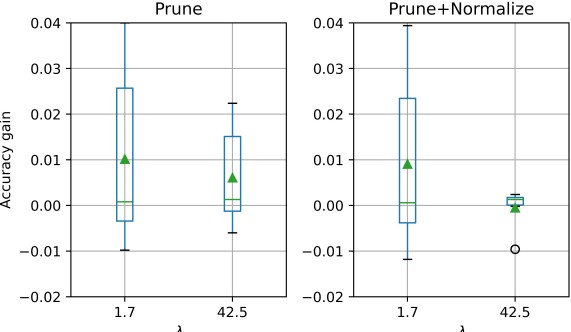

(a) Boxplots of ResNet18's accuracy gain compared to Original for different regularization strengths

(b) Boxplots of ResNet50's accuracy gain compared to Unedited model for different regularization strengths

Figure 6: Boxplots for the accuracy gain of Pruning and Pruning with Normalization over the Unedited mode for a given model and $\lambda$ value.

## F   Human-guided editing experiment - setup

To replicate the human-guided editing experiment, we used the following setup and design assumptions:

- We based our dataset on the same source as the Metashift experiment. The training dataset consisted of 100 samples of a class with spurious correlation, while the test dataset comprised 100 samples of the same class with correlations to any concepts except the one used in training.

- We utilized the same model and training methods as in the original repository with minor modifications (i.e., to the dataloaders and pruning capabilities) to align with initial results.

- Greedy (Oracle) and random pruning used the same number of simultaneous pruned concepts as the users. For this, we assumed that for each task, each unique number of concepts pruned by users results in a model (e.g., if users pruned between 1 and 6 concepts for a task, we would have one greedy/random model for 1 concept pruned, 2 concepts pruned and so on).

- We reproduced the user study following the details provided in the original paper by introducing the terminology to the users (Figure 8) and presenting a similar task of choosing concepts to prune per scenarios (Figure 9).

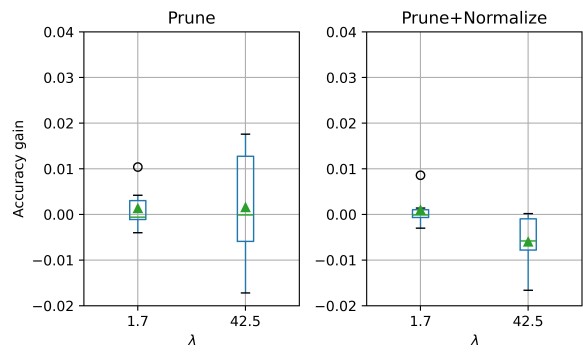

(a) Boxplots of Adam ResNet50's accuracy gain compared to Original for different regularization strengths

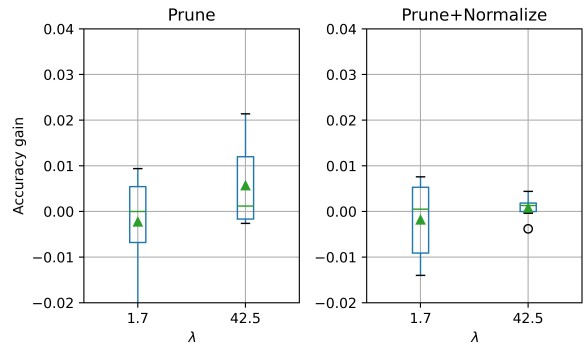

(b) Boxplots of Strict ResNet50's accuracy gain compared to Unedited model for different regularization strengths

Figure 7: Boxplots for the accuracy gain of Pruning and Pruning with Normalization over the Unedited mode for a given model and $\lambda$ value of other implementations.

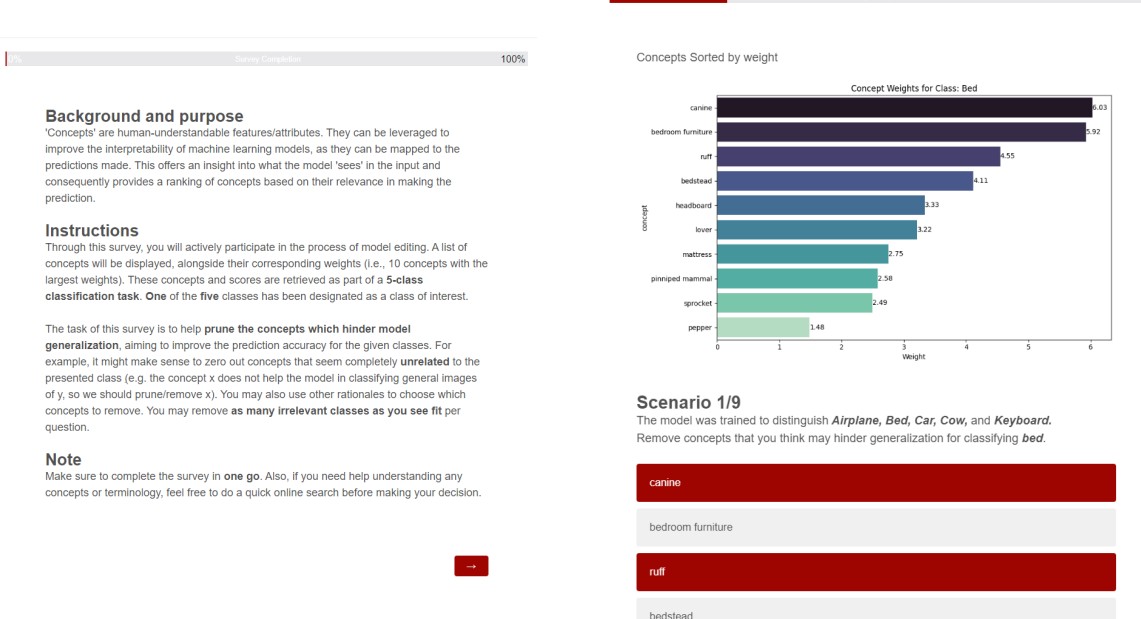

Figure 8: First page of the survey introducing the task.

Figure 9: Example scenario with choice of concepts

# G   User study for model editing

Replicating Table 8 of the original paper, Table 15 contains a full overview of the results obtained from the user study. The original authors used only the PCBM-h results for their breakdown. We attempt to replicate these results and while we also observe increases and decreases in performance across scenarios, the magnitude of the increases does not align with the original claim. Moreover, the usage of PCBM-h for reporting the final results does not provide a reliable insight into the method's performance. By visualizing the PCBM results instead (shown in Table 16), we can see that pruning (in the settings defined, where we highlight a larger number of simultaneous concepts pruned than the original paper) reduces the per-class accuracy considerably.

| Scenario | Unedited | Random Prune | User Prune | Greedy Prune |
|---|---|---|---|---|
| | Shifted Class Test Accuracy | | | |
| bed(dog) | $0.732 \pm 0.018$ | $0.727 \pm 0.019$ | $0.738 \pm 0.027$ | $0.729 \pm 0.021$ |
| keyboard(cat) | $0.918 \pm 0.003$ | $0.906 \pm 0.009$ | $0.903 \pm 0.010$ | $0.911 \pm 0.010$ |
| bed(cat) | $0.864 \pm 0.006$ | $0.863 \pm 0.008$ | $0.854 \pm 0.009$ | $0.911 \pm 0.010$ |
| couch(cat) | $0.965 \pm 0.005$ | $0.962 \pm 0.005$ | $0.961 \pm 0.007$ | $0.964 \pm 0.004$ |
| painting(lamp) | $0.534 \pm 0.021$ | $0.509 \pm 0.012$ | $0.509 \pm 0.012$ | $0.511 \pm 0.012$ |
| pillow(clock) | $0.764 \pm 0.010$ | $0.762 \pm 0.005$ | $0.759 \pm 0.007$ | $0.763 \pm 0.002$ |
| television(fireplace) | $0.701 \pm 0.012$ | $0.686 \pm 0.019$ | $0.678 \pm 0.015$ | $0.685 \pm 0.020$ |
| fork(tomato) | $0.565 \pm 0.015$ | $0.580 \pm 0.016$ | $0.579 \pm 0.014$ | $0.575 \pm 0.019$ |
| car(snow) | $0.618 \pm 0.008$ | $0.625 \pm 0.015$ | $0.620 \pm 0.016$ | $0.617 \pm 0.012$ |
| | Overall Test Accuracy | | | |
| bed(dog) | $0.903 \pm 0.002$ | $0.903 \pm 0.003$ | $0.905 \pm 0.006$ | $0.903 \pm 0.003$ |
| keyboard(cat) | $0.901 \pm 0.002$ | $0.898 \pm 0.002$ | $0.898 \pm 0.002$ | $0.899 \pm 0.002$ |
| bed(cat) | $0.853 \pm 0.002$ | $0.852 \pm 0.002$ | $0.851 \pm 0.002$ | $0.853 \pm 0.002$ |
| couch(cat) | $0.898 \pm 0.001$ | $0.898 \pm 0.002$ | $0.897 \pm 0.002$ | $0.898 \pm 0.002$ |
| painting(lamp) | $0.840 \pm 0.002$ | $0.840 \pm 0.002$ | $0.841 \pm 0.003$ | $0.841 \pm 0.002$ |
| pillow(clock) | $0.877 \pm 0.002$ | $0.879 \pm 0.002$ | $0.878 \pm 0.002$ | $0.879 \pm 0.002$ |
| television(fireplace) | $0.888 \pm 0.003$ | $0.886 \pm 0.004$ | $0.884 \pm 0.003$ | $0.886 \pm 0.005$ |
| fork(tomato) | $0.719 \pm 0.003$ | $0.726 \pm 0.004$ | $0.726 \pm 0.003$ | $0.725 \pm 0.004$ |
| car(snow) | $0.808 \pm 0.002$ | $0.809 \pm 0.003$ | $0.807 \pm 0.003$ | $0.807 \pm 0.003$ |

Table 15: Reproduction results of the user study pruning with PCBM-h.

## H  Multimodal performance based on initial choice of classes

In designing our user study experiment, we explored and considered different approaches with the dataset and multimodal representations. Notably, we had to consider the choice of initial elements (classes) used as a basis for utilizing CLIP and ConceptNet as it can significantly influence the results obtained.

When determining the classes for concept retrieval in our user study scenarios, we compared several approaches:

- Using the 5 main classes on which we are conducting classification.
- Incorporating these 5 classes along with the class exhibiting spurious correlations.
- Including all classes and spurious correlation classes used across the scenarios.
- Employing the CIFAR100 classes, which were part of the original code.

These different configurations allowed us to assess how class selection impacts the study's outcomes. Even with a 5-class simple classification task, the difference in accuracy (shown in Table 17) suggests that more classes/initial objects lead to better accuracy. Adding on to this, better interpretability of the data and predictions are showcased by the results with more suitable high-weighted concepts (for a class exhibiting "bed" with spurious context "cat" we see terms such as "bedroom furniture" and "feline" for CIFAR100 Figure 10d but in limited initial categories we see terminology from other classes spilling over such as "car seat" in Figure 10a).

| Scenario | Unedited | Random Prune | User Prune | Greedy Prune |
|---|---|---|---|---|
| | Shifted Class Test Accuracy | | | |
| bed(dog) | $0.614 \pm 0.166$ | $0.099 \pm 0.163$ | $0.051 \pm 0.107$ | $0.208 \pm 0.228$ |
| keyboard(cat) | $0.843 \pm 0.049$ | $0.199 \pm 0.274$ | $0.051 \pm 0.107$ | $0.236 \pm 0.293$ |
| bed(cat) | $0.886 \pm 0.029$ | $0.415 \pm 0.318$ | $0.239 \pm 0.279$ | $0.389 \pm 0.341$ |
| couch(cat) | $0.979 \pm 0.010$ | $0.390 \pm 0.370$ | $0.310 \pm 0.339$ | $0.461 \pm 0.385$ |
| painting(lamp) | $0.467 \pm 0.061$ | $0.132 \pm 0.143$ | $0.132 \pm 0.128$ | $0.188 \pm 0.165$ |
| pillow(clock) | $0.707 \pm 0.035$ | $0.264 \pm 0.234$ | $0.279 \pm 0.252$ | $0.364 \pm 0.271$ |
| television(fireplace) | $0.539 \pm 0.059$ | $0.117 \pm 0.167$ | $0.153 \pm 0.201$ | $0.219 \pm 0.225$ |
| fork(tomato) | $0.790 \pm 0.035$ | $0.174 \pm 0.229$ | $0.156 \pm 0.236$ | $0.216 \pm 0.269$ |
| car(snow) | $0.762 \pm 0.061$ | $0.175 \pm 0.024$ | $0.256 \pm 0.279$ | $0.224 \pm 0.269$ |
| | Overall Test Accuracy | | | |
| bed(dog) | $0.874 \pm 0.025$ | $0.774 \pm 0.030$ | $0.764 \pm 0.021$ | $0.795 \pm 0.044$ |
| keyboard(cat) | $0.839 \pm 0.012$ | $0.714 \pm 0.055$ | $0.718 \pm 0.043$ | $0.722 \pm 0.058$ |
| bed(cat) | $0.820 \pm 0.014$ | $0.735 \pm 0.064$ | $0.698 \pm 0.057$ | $0.739 \pm 0.068$ |
| couch(cat) | $0.852 \pm 0.010$ | $0.750 \pm 0.070$ | $0.736 \pm 0.065$ | $0.764 \pm 0.073$ |
| painting(lamp) | $0.806 \pm 0.013$ | $0.756 \pm 0.024$ | $0.758 \pm 0.021$ | $0.766 \pm 0.028$ |
| pillow(clock) | $0.854 \pm 0.005$ | $0.781 \pm 0.042$ | $0.784 \pm 0.045$ | $0.798 \pm 0.049$ |
| television(fireplace) | $0.825 \pm 0.009$ | $0.755 \pm 0.029$ | $0.762 \pm 0.034$ | $0.772 \pm 0.039$ |
| fork(tomato) | $0.738 \pm 0.012$ | $0.661 \pm 0.036$ | $0.661 \pm 0.034$ | $0.668 \pm 0.040$ |
| car(snow) | $0.800 \pm 0.007$ | $0.699 \pm 0.050$ | $0.714 \pm 0.054$ | $0.707 \pm 0.055$ |

Table 16: Performance of PCBMs in the context of user study pruning.

|  | Scenario classes | Scenario and Spurious classes | All Scenarios classes | CIFAR100 classes |
|---|---|---|---|---|
| Overall Test Accuracy | 0.797 ± 0.007 | 0.798 ± 0.015 | 0.802 ± 0.008 | 0.819 ± 0.014 |
| Shifted Class Test Accuracy | 0.864 ± 0.036 | 0.82 ± 0.051 | 0.852 ± 0.037 | 0.875 ± 0.024 |

Table 17: Top concepts retrieved based on the initial choice of classes.

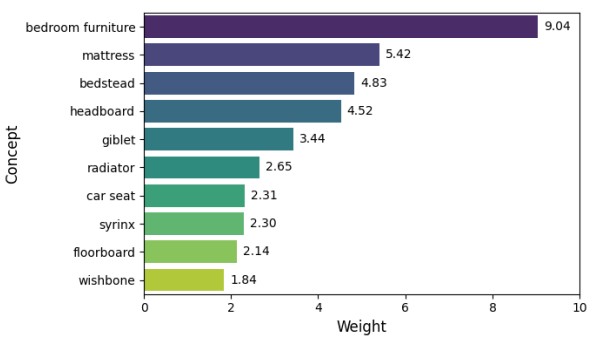

(a) Concepts extracted using the 5 scenario classes explained above.

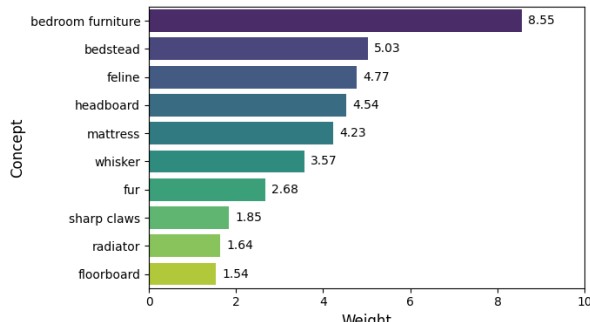

(b) Concepts extracted using the 5 scenario and spurious classes.

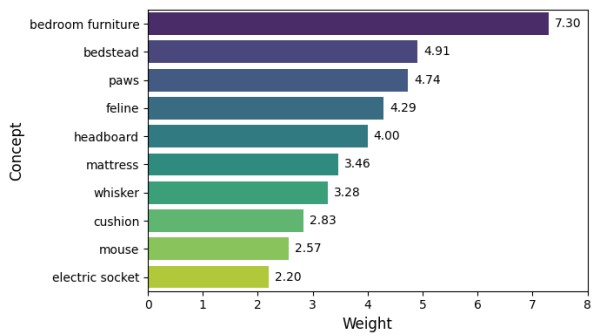

(c) Concepts extracted using all scenario classes.

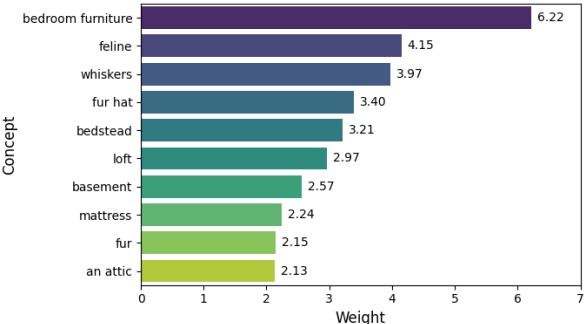

(d) Concepts extracted using CIFAR100 classes.

Figure 10: Concept extracted from the various scenarios established.

# I   Additional examples of obtained saliency maps

Below, we attach an example of the saliency maps for a television image. The concepts used are the four most important concepts for the class "television" and the concept floor which co-occurs with the television at the bottom of the image. The results obtained here align with the "bike" saliency map results.

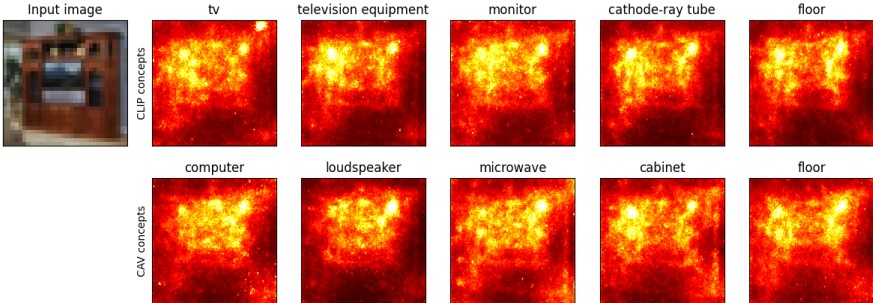

Figure 11: Extension results of the saliency map experiment. Shown here are ten saliency maps for an image from the class "television" of CIFAR100. Five CAV concepts and five CLIP concepts are used.

