# OpenReview forum: "[Re] On the Reproducibility of Post-Hoc Concept Bottleneck Models"
_TMLR — Accepted by TMLR_

### Review · Reviewer_JLhh · 2024-03-08

**Summary Of Contributions:**

In this paper, the authors reproduced Post-hoc Concept Bottleneck Models (PCBMs) to validate the experimental results presented in the original paper. This validation includes confirming whether PCBMs can 1) achieve performance similar to the original model, 2) operate without the need for labeled concept datasets, and 3) enable global model editing. Additionally, the interpretability and scalability of PCBMs were validated through a sanity check of concept vectors, assessment of alignment with actual concepts, and extension experiments into audio classification. Through these validations, this paper demonstrates that to some extent, the experimental results of previous literature can be reproduced, with the exception of global model editing. Additionally, the paper shows that PCBMs possess interpretability and scalability.

**Audience:**

Yes

**Broader Impact Concerns:**

There is no section for Broader Impact Statement. The significance of this research lies in verifying reproducibility of Post-hoc CBM and conducting its validation in various scenarios, under the assumption that Post-hoc CBM holds significance. It would be beneficial to include a Broader Impact Statement section based on these considerations.

**Claims And Evidence:**

No

**Requested Changes:**

1) In this paper, the authors compared the performances of their trained model with the reported performances in the original paper to initially verify if the performance is reproducible. To facilitate a proper comparison, it would be beneficial to include the performance reported in the original paper in the tables of the paper, allowing readers to compare effectively.


2) The interpretability aspect is crucial, and Section 3.5.2 touches on important points. However, there are considerations to be made regarding the following:

[a] mentions that CAVs are sensitive to negative samples, suggesting that issues may arise not only due to the method of projection but also inherent to CAV itself.
[b] demonstrates issues with concept association bias in vision-language models like CLIP, which could potentially enhance the discussion in Section 3.5.2.

[a] Kim, Siwon, et al. "Grounding Counterfactual Explanation of Image Classifiers to Textual Concept Space." Proceedings of the IEEE/CVF Conference on Computer Vision and Pattern Recognition. 2023.

[b] Tang, Yingtian, et al. "When are Lemons Purple? The Concept Association Bias of Vision-Language Models." Proceedings of the 2023 Conference on Empirical Methods in Natural Language Processing. 2023.

3) In Figure 1, the discrepancy between CLIP concepts and CAV concepts in textual representation begs further explanation. Why are they different? The observation that saliency maps highlight primary objects rather than fine-grained parts relevant to concepts is closely related to findings in various papers analyzing CLIP. Citing these papers could enrich the discussion.

4) Expanding the validation of audio classification would be beneficial. Currently, the paper only covers the use of multimodal models like AudioCLIP. Similar to CAVs, exploring the application of techniques like CAVs to audio data could provide a broader validation of Post-hoc CBM across different data domains. More examples are needed besides Figure 4, and the results presented in Section 4.2.3 only cover UrbanSound8K despite the use of various audio datasets mentioned in Section 3.5.3. Additionally, detailed explanations of the settings for PCBM-h (e.g., dimensions for the residual path) are lacking. Overall, there is a shortage of explanation and validation for these experiments.

5) Further minor comments include:

- Ensuring consistent terminology for the SIIM-ISIC dataset.
- Properly indicating in the main text when referring to figures and tables included in the appendix for readability.

**Strengths And Weaknesses:**

Strengths:

- The paper provides clear explanations of what is clear and unclear in order to reproduce the results.
- The validation of three scenarios to verify the operation of post-hoc CBM, beyond the results shown in the paper, is highly valid. Particularly, the validation of object-concept correspondence appears crucial.

Weaknesses:

- The analysis of experimental results is overly simplistic. Additional discussion seems necessary.
- Verification in other domains has been limited to performance aspects only, which is regrettable. More diverse examples and experimental validations are needed in this regard.

---

> ### Author Response · Authors · 2024-03-31
> **Comment Reply**
>
> Dear Reviewer JLhh,
>
> Thank you very much for your feedback! We appreciate the effort in providing valuable feedback on our paper.
>
> We have added our responses and action points based on your comments in the points below. Because of time
> constraints, we have decided to send this response a day early to further work on the revisions, which will be uploaded
> tomorrow. We appreciate your patience.
> - **[Including the performance reported in the original paper in the tables for effective comparison].**
> We have updated our tables to include the original results.
> - **[Considerations regarding the discussion].** Both papers indeed provide valuable insights about entanglement in both CLIP and CAV-based concept vectors. We will improve and adjust the current discussion of concept object correspondence using these insights.
> - **[The figure showing discrepancy between CLIP concepts and CAV concepts in textual representation].** The explanation relating to this has been added to the relevant section.
> - **[The observation that saliency maps highlight primary objects rather than fine-grained parts relevant to concepts is closely related to findings in various papers analyzing CLIP.]** We found two related papers, **[ref1]** finds that clip doesn’t perform well for region prediction, and hypothesize that this is because CLIP was trained to only match whole images to the corresponding text descriptions without capturing the fine-grained alignment between image regions and text spans. **[ref2]** looks specifically at saliency map methods used in combination with clip, they find that although it is possible to create a saliency map that only looks at the main object in a scene, most maps show saliency across more of the whole image. We have updated our discussion to reflect this.
>
>    ***[ref1]** Yiwu Zhong, Jianwei Yang, Pengchuan Zhang, Chunyuan Li, Noel Codella, Liunian Harold Li, Luowei Zhou, Xiyang Dai, Lu Yuan, Yin Li, and Jianfeng Gao. RegionCLIP: Region-based Language-Image Pretraining. http://arxiv.org/abs/2112.09106.*
>
>    ***[ref2]** Peijie Chen, Qi Li, Saad Biaz, Trung Bui, and Anh Nguyen. gScoreCAM: What Objects Is CLIP Looking At? https://link.springer.com/10.1007/978-3-031-26316-3_35. Series Title: Lecture Notes in Computer Science*
>
> - **[Validation of audio classification].** Originally, we only intended to use AudioSet and ESC-50’s labels as
> concepts for the classification of UrbanSound8K using AudioCLIP, but to further expand the validation of audio classification we have also added results for the actual ESC-50 dataset. We’ve also added more details regarding the settings for PCBM-h.
>
>    In addition, we could not find any concept annotated dataset for audio that can be directly utilized, meaning that experimentation with CAVs as the original authors intended would include the creation of a new densely annotated dataset which is unfortunately outside the scope of our paper. However, we’ve come up with a possible alternative to get CAVs and we are working on adding its results to Table 7. The approach can be briefly summarized as follows: For each label of our audio dataset, we find a set of words associated with it (via ConceptNet) that we’ll treat as concepts. Then for each concept generated, we gather a set of audios whose labels are associated with the concept and treat them as positive examples and the remainder as negative examples. We then proceed normally to get our concept vectors using SVMs to find the normal vector that separates the positive and negative examples for each concept.
>
>    We will compare the results obtained using this new approach with the CLIP based results on both the ESC-50 and US8k datasets, based both on the accuracy and the intuitiveness of their concepts. To facilitate this comparison we will extend Figure 4 to a larger table with more examples as proposed.
> - **[Ensuring consistent terminology for the SIIM-ISIC dataset].** We have adjusted this in the paper.
> - **[Properly indicating when referring to figures and tables included in the appendix].** This has also
> been adjusted in the paper.
> - **[Broader impact concerns].** We have expanded our conclusion (within the discussion section) to better
> highlight the significance of our work towards Post-hoc CBM. We agree that the importance of our research
> depends on the assumption that Post-hoc CBMs hold significance, and have thus added discussion relating to how researchers should be more careful when working with Post-hoc CBMs moving forward, especially given how frequently it has been cited/expanded upon. However, from our understanding, we believe that this would better fit as part of the discussion, due to how the broader impact statement generally focuses more on societal impacts rather than ones related specifically to the scientific community.
>
> Thanks again for your help in improving our paper, we truly appreciate it!
>
> Sincerely,
> The authors

---

### Review · Reviewer_VnXc · 2024-03-17

**Summary Of Contributions:**

_Note: To my understanding, this is a submission for the reproducibility challenge (MLRC) 2023. Please correct me if this is not._

This report aims to reproduce the results of Yuksekgonul et al. (2023), which proposed post-hoc concept bottleneck models (PCBM). The authors mainly focus on validating three core claims in the original paper. (1) PCBMs closely achieves the original model's performance. (2) PCBMs can be trained without labeled concept datasets. (3) PCBMs enable global model editing. This report successfully validates the first claim, but finds that the second claim can only be partially supported. The third claim, unfortunately, has not been reproduced well. Stepping further, the paper also conducts several additional experiments on PCBMs. One of the experiments, interestingly, suggest that the interpretabiliy of PCBMs may be more limited than it looks.

**Audience:**

Yes

**Broader Impact Concerns:**

I believe that the current version is already good in this respect.

**Claims And Evidence:**

Yes

**Requested Changes:**

The main issues that I have is about the clarity of the manuscript. In particular, I recommend:
- adding several sentences to the introduction that summarizes the findings---whether the results were satisfactory, what were the key differences, etc.
- slightly more explanations on the intentions behind the random projection experiment.
Other than these, I am generally happy with the manuscript.

**Strengths And Weaknesses:**

**Strengths**
- Authors have successfully identified and distilled the key claims of the original paper, and systematically validated the claims through experiments. In particular, the categorization of the main points into three claims have been very effective and added clarity to the overall manuscript.
- The paper has tediously reported small experimental details, even including the hardware details, for future references.
- The paper accurately points out the details that have been missing in the original paper, and clearly explains how the authors approached to fill the gap.

**Weaknesses**
- Writing could have been a bit more clearer. I could not locate the main messages of the manuscript very easily. In many cases, the key sentences appeared in the middle of a lengthy section or a paragraph, instead the beginning.
- The motivation behind the additional "random projection" experiment was not very clear to me. Is this merely a sanity check that the concept matrix actually plays a significant role in PCBMs? Also, it is not super clear to me why the authors brought up the J-L lemma; it adds some mathematical fanciness, but I do not think the lemma adds any concreteness to the discussion.
- The choice to reduce the COCO-Stuff and SIIM-ISIC datasets to a smaller scale is totally understandable, but nevertheless undermines the credibility of the reproducibility study.

---

> ### Author Response · Authors · 2024-03-31
> **Comment Reply**
>
> Dear Reviewer VnXc,
>
> Thank you very much for your feedback! We appreciate the effort in providing helpful feedback on our paper. Because of time constraints, we have decided to send this response a day early to further work on the revisions, which will be uploaded tomorrow. We appreciate your patience.
>
> We will respond to each of your requested changes and comments below:
>
> - **[Clarity of writing].** We have refactored the writing such that shorter sentences are used which should better convey the message being covered.
>
> - **[The motivation behind "random projection"].** The experiment is meant to create a baseline for CLIP and CAV concepts. If the performance of PCBMs with CLIP and CAV concepts is lower than with random concepts it would raise questions about the validity of PCBMs. However, since PCBMs perform worse with random concepts, we designed more insightful experiments to examine their validity. This means random projections were important for our understanding of PCBMs, but hold less value in the context of the finished paper.
>
>     The main goal of the J-L lemma was to explain the poor performance seen for the HAM10000 dataset in Table 5. Together with this poor performance, the J-L lemma tells us that when the number of concepts is small enough we can worry "less" about the interpretability of our concepts. Since the concept vectors at least hold more information than random. However truly formalizing the meaning of the previous two sentences is quite difficult, which is why the section in the paper is vague. We think it would be best to reduce the section about random projections and focus on its relevance as a baseline.
>
>  - **[The reduction of the COCO-Stuff and SIIM-ISIC datasets].** We believe there is a small misunderstanding here; we replicate the original authors' own preprocessing pipeline which includes sampling the original datasets. Our contribution here is to provide these preprocessed datasets and the scripts to create these for transparency and reproducibility. We have adjusted this for more clarity in the updated manuscript.
>
> Again, we truly appreciate the review and are open to any further suggestions you may have!
>
> Sincerely,
>
>
> The authors

---

### Review · Reviewer_XbjT · 2024-03-19

**Summary Of Contributions:**

The work is a reproduction effort of [1]. As such, it

-  checks the reproducability of experimental results in support of [1]'s claims using [1]'s released codes as well as own investigation and implementation for the missing parts.
-  conducts an independent user study mostly using the conditions of [1]'s study.
-  investigates the importance and validity of concepts in concept-bottleneck interpretability of [1] through an ablation study of generated concepts as well as a semantic evaluation of the discovered concepts.
-  evaluates the method on audio data that is beyond [1]'s evaluation setup
-  improves codebase to be more user-friendly

[1] Yuksekgonul, Mert, Maggie Wang, and James Zou. "Post-hoc Concept Bottleneck Models." The Eleventh International Conference on Learning Representations. 2022.

**Audience:**

Yes

**Claims And Evidence:**

Yes

**Requested Changes:**

Are the reported improvements in Table 3 (“Edit Gain”) statistically significant? Can this be repeated to make sure the differences, especially in case of pruning, is reliable?

- p3. <.,.> should be defined as the dot product.
- eq1. closing parenthesis for the loss should come after label y.
- eq1. sampling (~) or membership ($\in$) are usually not represented by the minus sign (-)
- p2. if $\mathbf{\omega}$ is a vector, then g() can only represent a binary classifier, is that the intention?
- eq2. parentheses need to be corrected
- check when the differences are meant to be percentage points and use percentage points in the text instead of percent (%).
- claim 2 discussions in p.8 do not refer to the relevant table. Related to that, it is better to have the results in Table 1 repeated in Table 2 and named differently to compare with and without provided concepts.
- similarly, the results for the userstudy in Claim 3 do not refer to the corresponding tables.
- for Table 5. good to have the results from “meaningful” concepts repeated here for comparison.

**Strengths And Weaknesses:**

**Strengths.**

-it identifies the missing information required for reproduction and makes a plausible effort either by looking in the authors’ codebase or prior work to fill in the gap, implement, and experiment accordingly.
- the list of required assumptions corresponding to the missing information is further provided.
- addition of a new benchmark of a different modality (audio) is interesting
- the conduction of a new user study helps with understanding the reliability of the original work.
- there are informative findings regarding the reproducibility of [1]
    - the main claim, regarding the matching performance, is generally reproducible, including on the new audio modality
    - underperformance of the reported CUB baseline in [1]
   - smaller improvement using manual editing of concepts.
   - inexisting improvement and sometimes degradation of results with the userstudy, especially when it comes to non-hybrid PCBM.

**Weaknesses.**

- presentation can be significantly improved, some examples in the requested changes.
- it is not clear how the use of “random projection” for the concepts is relevant for the PCBM study.
- it is not clear how the object-concept correspondence values in Table 6 are produced.
- it is not clear how the concepts are formed for the audio classification experiments from section 4.2.3.

---

> ### Author Response · Authors · 2024-03-31
> **Comment Reply**
>
> Dear Reviewer XbjT,
>
> Thank you very much for your feedback and acknowledgement of our paper's strengths! We appreciate the valuable feedback provided on our paper. Because of time constraints, we have decided to send this response a day early to further work on the revisions, which will be uploaded tomorrow. We appreciate your patience.
>
> Below you can find our responses to each of your requested updates and remarks:
>
> -**[Clarity of presentation].** We have adjusted the writing based on the requested changes provided.
>    1. $<.,.>$ on page 3 is now defined as the dot product.
>
>    2. The closing parenthesis for the loss now comes after label y in eq. 1. We have also changed the minus sign to $\sim$.
>    3. $g()$ in page 2 is indeed supposed to represent both binary and multivariate classifiers, thus **w** should be **W** and $b$ should be a vector **b**.
>    4. Equation 2 should indeed be: $$\underset{r}{min} \sim \mathbb{E}_{(x,y) \sim D}[\mathcal{L}(g(f_C(x)) +r(f(x)),y)]$$
>    5. We have adjusted the use of percentage points accordingly in-text.
>    6. Claim 2 in discussions (page 8) now should refer to the relevant table. Also, we now have the results in Table 1 repeated in Table 2 and renamed them appropriately.
>    7. Claim 3 also now refers to the correct results.
>    8. Table 5 now repeats the results from “meaningful” concepts for better comparison.
>
> -**[The motivation behind "random projection"].**  The experiment is meant to create a baseline for CLIP and CAV concepts. If the performance of PCBMs with CLIP and CAV concepts is lower than with random concepts it would raise questions about the validity of PCBMs. However, since PCBMs perform worse with random concepts, we designed more insightful experiments to examine their validity. This means random projections were important for our understanding of PCBMs, but hold less value in the context of the finished paper. We think it would be best to reduce the section about random projections and focus on its relevance as a baseline.
>
> -**[How the object-concept correspondence values are produced].** We have adjusted the relevant explanations to make this clearer.
>
> -**[The concept formation for the audio classification experiments].** This has also been adjusted to better convey how these concepts were gathered and compiled.
>
> Again, thank you very much for the thorough review. We truly appreciate it!
>
> Sincerely,
>
> The authors

---

### Author Response · Authors · 2024-04-01
**Revised version of submission**

Dear reviewers,

We’ve incorporated all your feedback in this new final revision.
The improvements are mainly in terms of clarity of writing, additional experiments and the discussion of results.
To facilitate the requested changes we have had to extend the manuscript to 13 pages of main content.

Sincerely,

The authors

---

### Decision · Action_Editor_LBu8 · 2024-04-26

**Recommendation:** Accept as is

**Comment:**

This paper is reviewed by three expert reviewers. During the rebuttal period, the authors addressed the concerns raised by the reviewers well.
After the rebuttal, all the reviewers were willing to accept the paper.

Following the TMLR official acceptance criteria (https://jmlr.org/tmlr/acceptance-criteria.html), I mainly focused on whether this paper has sufficient generalized insights and actionable lessons for the TMLR audience.

Note that all the experiments of this reproducibility report are based on the authors' code repository. I don't think showing the re-run results of the original code repository itself is sufficient generalized insights. Thus, my decision is based on the additional contributions of the authors. Namely, random projection experiments, object-concept correspondence, and audio classifier results. After carefully reading the reviewers' comments and the revised paper, I think the contribution of this paper is slightly above the bar of the TMLR criterion.

Specifically, I think the conclusion of this paper would be somewhat interesting for the audiences interested in applying PCBM (or any other CBM variants) to their own task rather than the well-known benchmarks (such as CUB). We still have a large gap for improving CBMs in terms of its interpretability and generalizability.

**Audience:**

This paper would be interesting to the audiences working on CBM but in different domains rather than the well-known benchmarks, such as CUB.

**Claims And Evidence:**

This paper is a reproducibility report of the Post-Hoc Concept Bottleneck Model (PCBM) (Yuksekgonul et al., 2023). This paper includes (1) reproducing the experimental results and numbers, (2) reproducing the user study, (3) new experimental results with random projection, (4) new experimental results for showing object-concept correspondence, and (5) new audio dataset results.

This study shows that the three original claims (PCBMs achieve comparable performance to the original model, PCBMs do not require labeled concept datasets, and PCBMs allow for global model editing) are partially reproducible; the first two claims are reproducible, but the last one shows a somewhat mixed result. Also, this study shows that when using PCBM to audio datasets, the results become mixed again.